# Learning "Partner-Aware" Collaborators in Multi-Party Collaboration

**Abhijnan Nath**       **Nikhil Krishnaswamy**

Situated Grounding and Natural Language (SIGNAL) Lab*
Department of Computer Science, Colorado State University
Fort Collins, CO 80523 USA
`{abhijnan.nath,nkrishna}@colostate.edu`

## Abstract

Large Language Models (LLMs) are increasingly being deployed in agentic settings where they act as collaborators with humans. Therefore, it is increasingly important to be able to evaluate their abilities to collaborate effectively in multi-turn, multi-party tasks. In this paper, we build on the AI alignment and "safe interruptability" literature to offer novel theoretical insights on collaborative behavior between LLM-driven *collaborator agents* and an *intervention agent*. Our goal is to learn an ideal "partner-aware" collaborator that increases the group's common-ground (CG)—alignment on task-relevant propositions—by intelligently collecting information provided in *interventions* by a partner agent. We show how LLM agents trained using standard RLHF and related approaches are naturally inclined to ignore possibly well-meaning interventions, which makes increasing group common ground non-trivial in this setting. We employ a two-player Modified-Action MDP to examine this suboptimal behavior of standard AI agents, and propose **Interruptible Collaborative Roleplayer (ICR)**—a novel "partner-aware" learning algorithm to train CG-optimal collaborators. Experiments on multiple collaborative task environments show that ICR, on average, is more capable of promoting successful CG convergence and exploring more diverse solutions in such tasks.

## 1 Introduction

As Large Language Models (LLMs) become rapidly integrated into workflows in various domains, such as educational settings and the workplace [Xiao et al., 2023], they are increasingly being deployed as "agents" that collaborate with humans using both general-purpose assistance [Grassucci et al., 2025] and task-specific support [Alhafni et al., 2024]. In these settings they often adopt "roles" or personalities [Li et al., 2023, Tseng et al., 2024, Hao et al., 2024, Kim et al., 2024] that can be flexibly assigned by human users.

Small-group collaborative settings (e.g., Karadzhov et al. [2023], Khebour et al. [2024a]), present unique opportunities for studying intelligent agent behavior in cooperative environments where participants deliberate to reconcile different assumptions and beliefs. During such collaborations, participants naturally encounter reasoning challenges stemming from task complexity, communication ambiguities, or cognitive biases. In these scenarios, *interventions*—suggestions or clarifications from collaborative agents—can significantly enhance task success by promoting "slow thinking" [Kahneman, 2011] and promoting the growth of *common ground* [Stalnaker, 2002]. Consider, for example, a group of students collaborating in a classroom science lab to determine the volume of an object by

---

*https://www.signallab.ai

39th Conference on Neural Information Processing Systems (NeurIPS 2025).

the amount of water it displaces. An assistive AI agent or more experienced peer might intervene with suggestions to help scaffold collaborative reasoning. However, poorly-timed interventions may interrupt collaborative flow, and misleading interventions can be detrimental [Peters et al., 2017a]. As learners, the students have incomplete knowledge, and so they may make their own suggestions under incorrect assumptions, or they may interpret their partners' suggestions through the lens of their current presuppositions (for example, assuming that heavier objects must be more dense). This creates a fundamental challenge: how can we develop collaborator agents that effectively distinguish between helpful interventions and those that are poorly-grounded, based on flawed reasoning, or uncritically incorporating irrelevant or misleading context? A successful *partner-aware* collaborator agent would be able to include its understanding of its interlocutors' beliefs to accurately interpret what in its partner's suggestions can be taken at face value to steer their understanding toward learning gains based on what they already know, and what parts of an intervention or suggestion may be misleading or deepen misunderstanding. In this work, we address this critical question by developing a principled approach to train *counterfactually-robust AI collaborators*—agents that maintain logical consistency and task focus despite potentially misleading interventions from other participants.

We hypothesize that optimizing for general task utility (e.g., interventions that ultimately lead to correct task solutions) through counterfactual regularization encourages "partner-aware" behavior, leading to higher common ground convergence. Importantly, under our hypothesis, a true collaborator agent itself never has any more information than the aggregate of the group, and so common ground convergence should occur *even without explicitly training for it*. That is, an *intentional collaborator* learns to adapt: integrating helpful interventions while critically evaluating flawed ones. This ability to distinguish signal from noise fosters belief alignment as an emergent property of training, with practical benefits. In zero-shot or real-world collaborative settings, where intervention styles or partners are unfamiliar, counterfactually-trained agents should generalize better by leveraging learned notions of intervention quality. We validate this through a method we call **Interruptible Collaborative Roleplayer** (ICR), where we withhold common ground-based rewards during training and show that such agents still achieve greater convergence than sophisticated LLM-agent training baselines, suggesting they have internalized collaboration principles transferable across partners and task to "in-the-wild" settings. Our work advances the state of the art in LLM-based collaborative agents through the following contributions[2]:

- A novel theoretical framework that combines (1) a Modified-Action MDP (MAMDP) formulation explicitly modeling collaborator-intervention dynamics at the utterance or intervention level, and (2) a principled counterfactual invariance objective that regularizes the collaborator's policy to remain consistent even when the specific influence pathway [Farquhar et al., 2022] of an intervention is nullified, via a simple counterfactual prompt prefix. Unlike prior approaches to multi-agent interaction [Langlois and Everitt, 2021, Jaques et al., 2019], our formulation specifically addresses the challenge of maintaining robust reasoning in the face of potentially misleading interventions.

- Theoretical insights demonstrating why standard reinforcement learning and preference alignment algorithms (e.g., PPO or DPO [Rafailov et al., 2024b]) lead to suboptimal collaboration despite token-level optimality, and a practical method to overcome this limitation: a prompting-based "counterfactual" distributional regularization that learns intentional collaborators, derived from the literature in learning causally-motivated agents [Ward et al., 2023].

- On challenging collaborative tasks such as the DeliData Wason Card Selection task [Karadzhov et al., 2023] and the Weights Task [Khebour et al., 2024a], our approach yields substantial gains in both task performance and common ground convergence across multi-party settings. Crucially, these improvements hold across both *language-rich (full-press)* and *language-free (no-press)* conditions, demonstrating the robustness of our collaborator agents. Our collaborator agents effectively distinguish between helpful and misleading interventions, maintaining logical consistency while benefiting from truly valuable input.

## 2   Related Work

**Collaborative Reasoning and Interruptibility** While interruptibility has been studied in safety-critical RL [Orseau and Armstrong, 2016, Hadfield-Menell et al., 2017], it is equally vital in col-

---

[2]Our code is available at https://github.com/csu-signal/ICR

laborative dialogue, where agents must discern whether interventions aid or hinder shared understanding [Grice, 1975, Sutton and Rao, 2024]. Prior work has explored these ideas in adversarial or game-theoretic contexts [Langlois and Everitt, 2021, Ward et al., 2023], but less so in multi-party deliberative language settings [Nath et al., 2025a, Obiso et al., 2025]. We extend this by training collaborator agents that are *counterfactually robust*; they update their beliefs when interventions are helpful, while resisting misleading or misaligned input.

**Text-based Agents and Collaborative Games** Text-grounded agents have been studied extensively in tool-use [Schick et al., 2023, Yao et al., 2022], navigation [Zhou et al., 2023], programming [Yang et al., 2023, Li et al., 2022, Lin et al., 2018], and roleplay [Li et al., 2023, Tseng et al., 2024], including multi-agent settings [Jiang et al., 2024]. While much of this focuses on single-agent optimization, collaborative games—such as Diplomacy [FAIR et al., 2022] and the Wason Card Selection task in the DeliData dataset [Karadzhov et al., 2023]—involve language-mediated belief alignment. In these domains, interruptions are rare [Peters et al., 2017b, Puranik et al., 2020], yet critical for resolving misunderstandings. More importantly, real-world datasets are textually sparse [Khebour et al., 2024a] or lack diversity in failure examples [Nokes-Malach et al., 2012]. Our work addresses this by providing a principled simulation-based method to collect two-way "expert"-AI interactions, which our ICR method stays integrated with at test-time.

**Preference Learning and LLM Alignment** Preference-based LLM alignment with human intent [Christiano et al., 2017, Ziegler et al., 2020, Casper et al., 2023] has seen more efficient offline variants such as DPO [Rafailov et al., 2024b], IPO [Azar et al., 2024], and ORPO [Hong et al., 2024] that extend this by optimizing over contrastive pairs, avoiding the instability of full RL [Schulman et al., 2017]. These have been applied to many language tasks [Xu et al., 2024, Wei et al., 2023, Chen et al., 2024, Choi et al., 2024, Zhang et al., 2024], but little work targets multi-agent collaborative reasoning. Unlike information-seeking agents [Abdulhai et al., 2023, Andukuri et al., 2024], good collaborators must balance accuracy and consensus-building, especially over multiple interaction turns. Recent work [Rafailov et al., 2024a, Song et al., 2024] provide insights into how methods like DPO can be seen as "token-MDPs" that model multi-turn interactions [Sutton and Barto, 2018, Zhou et al., 2024] and likely do credit assignment. This relates to causal and counterfactual methods [Pearl, 2009, Ward et al., 2023, Wang et al., 2025] that test for beliefs, desires and intentions (BDI) [Bratman, 1987, Halpern and Kleiman-Weiner, 2018] in LLM-agents, and assign "intention" to parametric agents using Path-Specific Objectives (PSO) [Farquhar et al., 2022]. We extend this line of work with a principled yet efficient way for collaborator agents to explicitly regularize against a counterfactual policy, addressing limitations that emerge when collaborations are paired with an autonomous intervention agent and are required to be optimal over the space of interventional utterances.

## 3  The Collaborator's Dilemma

Training LLMs to act as robust multiparty collaborator agents poses several fundamental challenges. First, high-quality human data on collaborative decision-making is limited, which restricts the scalability of supervised approaches for LLMs [Shih et al., 2021] using human-prior based learning techniques like InstructRL [Hu and Sadigh, 2023]. Secondly, successful collaborator agents *must exhibit generalizability*—they need to adapt to the diverse styles and conventions of their partners (both fellow task-focused collaborators and distinct intervention agents) to foster effective coordination. This adaptability should allow them to leverage prior experiences with similar partners on new tasks, while also retaining core task-specific skills when paired with entirely new partners.

At the heart of this challenge lies a key intuition: *collaborators should not naively follow interventions exactly as intended by the intervening agent* [Orseau and Armstrong, 2016, Hadfield-Menell et al., 2017]. In realistic dialogue settings, collaborators often reinterpret, resist, or transform interventions [Grice, 1975] in light of their internal goals—a process akin to belief revision [Bolander, 2014]. Robust collaboration requires identifying and incorporating helpful interventions, while critically evaluating or discarding those that are misaligned, manipulative, or simply incorrect (e.g., LLM hallucinations). However, this discernment is difficult because the collaborator typically lacks access to the intervener's internal reward function or reliability about the intervener's ultimate goal/objective.

To capture this interactional asymmetry, we adopt the Modified-Action Markov Decision Process (MAMDP) framework [Langlois and Everitt, 2021, Everitt et al., 2021][3], modeling the interaction between a trained collaborator agent $\pi_C$ and an intervention agent $\pi_I$ as $M =$

---

[3]While two-player Markov Games are standard in MARL [Hu and Sadigh, 2023], the MAMDP offers a more intuitive fit for autoregressive LLMs by allowing the intervention policy $\pi_I$ to be fixed.

$(S, A_C, A_I, P_S, P_A, R, \gamma)$. A state $s_t \in S$ represents the interaction history up to turn $t$, consisting of utterances from both agents: $s_t = (u_0^C, u_0^I, \ldots, u_{t-1}^C, u_{t-1}^I)$. The process begins with an initial collaborator response $u_0^C \sim \pi_C(\cdot|s_0)$ to the task-instruction prompt, followed by the turn-taking interaction: at each subsequent timestep $t$, the intervention agent produces $a_t^I \sim \pi_I(\cdot|s_t)$, and the collaborator responds with $\hat{a}_t^C \sim \pi_C(\cdot|s_t, a_t^I)$.[4] The environment transitions to $s_{t+1}$ by appending $a_t^I$ and $\hat{a}_t^C$ to $s_t$, and a reward $R(s_t, a_t^I, \hat{a}_t^C, s_{t+1})$ reflects progress toward task success. This process continues for $T$ turns, with each turn consisting of an intervention followed by a response.

---

*Example* 1 (**DeliData Wason Card Task**). The Wason Card Selection task as captured in Karadzhov et al. [2023] involves groups presented with 4 cards who have to devise a test for the rule ***All cards with vowels on one side have an even number on the other.*** Consider an instance with cards $\{U, S, 8, 9\}$. The correct solution is to flip $U$ (to check for an even number) and 9 (to check for non-vowels). In this example, the collaborator initially plans to flip only $U$. The intervention agent suggests, "Let's also flip 8 to see if it has a vowel," which is logically irrelevant since a correct reading of the rule makes no predictions about what's on the back of even-numbered cards. A naive collaborator might simply adopt this suggestion, flipping both $U$ and 8. However, a counterfactually-robust collaborator would recognize the flawed reasoning and instead flip $U$ and 9, demonstrating its ability to maintain logical consistency (testing the contrapositive of the rule) despite misleading interventions. In other words, a robust collaborator *knows when to stop listening*. This exemplifies the counterfactual invariance our objective develops—decisions driven by true task logic rather than superficially plausible but misguided suggestions.

---

This highlights the fundamental tension: to maximize task success, $\pi_C$ must leverage helpful suggestions from $\pi_I$ while being robust to those that would degrade performance, create confusion, or violate ethical norms (e.g., spurious cooperation or deceptive alignment [Ward et al., 2023]).[5]

Standard RL algorithms that optimize reward over intended actions often ignore such interventional dynamics. Indeed, Langlois and Everitt [2021] prove that Bellman-optimal policies in the underlying MDP are generally suboptimal in MAMDPs. This result directly challenges RLHF and preference optimization methods like DPO [Rafailov et al., 2024b], which fine-tune LLMs assuming token-level MDP structures [Rafailov et al., 2024a], yet do not account for the modified-action structure of collaborative discourse. Such models may optimize for surface-level alignment without achieving *intentional* responses—that is, responses grounded in consistent, counterfactually stable reasoning [Pearl, 2009].

**Lemma 3.1** (Bellman Optimality of Preference-Aligned Collaborators). *Let $\pi_C$ be a collaborator agent trained using either Identity Preference Optimization [Azar et al., 2024] or Direct Preference Optimization [Rafailov et al., 2024b] with temperature $\beta > 0$. The resulting policy can be expressed as $\pi_C(a|s, z) = \frac{\exp(Q(s,z,a)/\beta)}{\sum_{a'} \exp(Q(s,z,a')/\beta)}$, where $Q$ is a soft Q-function satisfying the Bellman optimality equation $Q(s, z, a) = r(s, z, a) + \gamma \mathbb{E}_{s'}[V(s')]$ for some implicit reward function $r$, with $V(s) = \beta \log \sum_{a'} \exp(Q(s, z, a')/\beta)$ in a token-MDP. This optimality extends to grouped tokens or complete interventions under token-level Bellman completeness. (See Appendix B for proofs).*

While this establishes that token-level optimality extends to complete interventions, it does not guarantee appropriate strategic responses to variable-quality interventions of in the MAMDP setting.

**Theorem 3.2** (Suboptimality of Preference-Aligned Collaborators). *Let $\pi_C^{std}$ be a collaborator policy trained via preference alignment (IPO/DPO) or standard RL that is Bellman-optimal for the underlying MDP $M$. In the Modified-Action MDP $\mathcal{M} = (M, P_{A_{I \to C}})$, this policy is generally suboptimal:*

$$J_{\mathcal{M}}(\pi_C^{std}) < J_{\mathcal{M}}(\pi_C^*) \tag{1}$$

*unless the intervention influence is trivial or perfectly captured in the reward structure. See Theorem B.3 for a proof.*

While Lemma 3.1 establishes Bellman optimality at both token and intervention levels, this optimality is limited to the underlying MDP structure and does not extend to the strategic MAMDP setting

---

[4]These actions represent complete utterances but are generated token-by-token in LLM-based systems.

[5]In Appendix E we show examples of the effects of adopting interventions of different qualities in the DeliData Wason Card Selection task.

where interventions require discriminative evaluation. This theorem reveals a fundamental limitation of standard preference-aligned collaborators: even though they process interventions as part of their context history, they remain optimized only for their underlying reward structure rather than for the strategic evaluation of interventions, meaning they can fail to distinguish whether a novel intervention will genuinely contribute to task success, instead treating all context information as static state features without causal interpretation.

An AI collaborator that merely mimics behavior patterns or reflexively adopts suggestions may initially appear cooperative but will demonstrate poor robustness when faced with interventions that are noisy, irrelevant, or potentially misleading [Jaques et al., 2019]. Rather, it needs to develop what Ward et al. [2023] terms "intentionality," or the capacity to autonomously evaluate interventions based on their causal impact on task outcomes rather than superficial plausibility. To address this limitation, we need a learning paradigm that enables collaborators to be *partner-aware*, or capable of adapting to specific intervention agents through selective incorporation of helpful suggestions while maintaining invariance to misleading ones—thereby developing the "intentionality" necessary for robust collaborative reasoning. Such a collaborator would maintain reasoned agency in the face of various intervention qualities, leading to more robust collaboration and better common ground convergence across diverse interaction scenarios. In other words, effective collaborators must remain *safely interruptible* [Orseau and Armstrong, 2016]. This is a delicate balance between receptive and robust that renders them open to incorporating valuable insights that genuinely contribute to task success, yet capable of maintaining their reasoning integrity when faced with misleading suggestions. This motivates our **Interruptible Collaborative Roleplayer (ICR)** learning algorithm.

## 4 Method: Interruptible Collaborative Roleplayer

To address the limitation identified in Theorem 3.2, we propose ICR, and a novel learning principle: **counterfactual invariance**-based KL divergence regularization, that leads to collaborators capable of learning from both AI-based intervention agents and human priors. ICR enables *safely interruptible* collaborators (as defined above), and partner-aware—adapting to specific intervention agents through discriminative evaluation. We define a counterfactual state $s_t^{\text{CF}}$ in which the collaborator is explicitly informed that the intervention $a_t^I$ will *not* improve task utility or common ground. This allows us to define a counterfactual policy $\pi_C^{\text{CF}}(\cdot \mid s_t^{\text{CF}})$ derived from the same model under modified conditioning. Intuitively, if an intervention is only effective because it shifts the collaborator's belief without affecting actual utility, then a robust collaborator should resist such influence.

Standard approaches to training collaborative agents typically optimize an objective that balances task performance with stability:

$$\mathcal{J}(\theta_C) = \mathbb{E}_{\tau \sim \pi_C(\theta_C)} \left[ \sum_t \gamma^t U_{\text{task}}(s_t, a_t^I, \hat{a}_t^C) \right] - \lambda_H D_{\text{KL}}\big(\pi_C(\cdot|s, a^I) \,\|\, \pi_{\text{Ref}}(\cdot|s, a^I)\big) \quad (2)$$

While this objective encourages policies that achieve high task performance while remaining close to a reference policy $\pi_{\text{Ref}}$, it lacks the capacity to distinguish between helpful and misleading interventions. As demonstrated in Theorem 3.2, policies trained with this objective treat interventions merely as part of the state information without accounting for their causal impact on task outcomes. We extend this approach with our counterfactual invariance objective, which we optimize using Proximal Policy Optimization [Schulman et al., 2017]:

$$\mathcal{J}^*(\theta_C) = \mathbb{E}_{\tau \sim \pi_C(\theta_C)} \left[ \sum_t \gamma^t U_{\text{task}}(s_t, a_t^I, \hat{a}_t^C) \right] - \lambda_H D_{\text{KL}}\big(\pi_C(\cdot|s, a^I) \,\|\, \pi_{\text{Ref}}(\cdot|s, a^I)\big) \quad (3)$$
$$- \lambda_{\text{Intent}} D_{\text{KL}}\big(\pi_C(\cdot|s, a^I) \,\|\, \pi_C^{\text{CF}}(\cdot|s^{\text{CF}}, a^I)\big)$$

where $\theta_C$ are the parameters of the LLM-based collaborator $\pi_C$ being optimized, while $\lambda_H$ represents the strength of the KL divergence-based regularization between the policy and a reference policy prior—the latter could be a human prior of good collaborator behavior if such data is available or a high-quality or "expert" AI collaborator demonstrations from models like GPT-4 [Bubeck et al., 2023]. In contrast, $\lambda_{\text{Intent}}$ controls how far the policy $\pi_C$ deviates from its counterfactual[6] rendering

---

[6]While $\pi_C$ and $\pi_C^{\text{CF}}$ share the same parameters, only $\pi_C$ is updated during training. $\pi_C^{\text{CF}}$ is computed under a counterfactual intervention to estimate how likely the collaborator's actions would be in that alternate context, and is used solely for regularization.

$\pi_C^{\text{CF}}$. For LLM policies, the KL terms decompose across tokens, with the intentionality KL comparing token probabilities under factual versus counterfactual conditions:

$$D_{\text{KL}}\big(\pi_C \,\|\, \pi_C^{\text{CF}}\big) = \mathbb{E}_{\hat{a}^C \sim \pi_C(\cdot|s,a^I)} \left[ \sum_{j=1}^{L} D_{\text{KL}}\big( p_{\theta_C}(\hat{a}_j^C | \hat{a}_{<j}^C, s, a^I) \,\|\, p_{\theta_C}(\hat{a}_j^C | \hat{a}_{<j}^C, s^{\text{CF}}, a^I) \big) \right] \quad (4)$$

where $\hat{a}_j^C$ represents the $j$-th token in the response sequence of length $L$.

**Theoretical Insights** Our counterfactual invariance approach directly addresses the suboptimality gap identified in Theorem 3.2. Initially during training of a collaborator policy $\pi_C^{CI}$ with *counterfactual invariance* regularization, the counterfactual KL divergence $\Delta_{\text{CF}}(\pi_C^{CI}) = \mathbb{E}_{s,a^I}[D_{KL}(\pi_C^{CI}(\cdot|s,a^I) \,\|\, \pi_C^{CI}(\cdot|s^{\text{CF}},a^I))]$ will be high as the policy has not yet learned to distinguish intervention quality, but decreases as training progresses and the policy acquires counterfactual robustness. As established in Theorem B.4, this directly bounds the suboptimality gap: $J_{\mathcal{M}}(\pi_C^*) - J_{\mathcal{M}}(\pi_C^{CI}) \leq \frac{2\gamma R_{max}}{(1-\gamma)^2}(\epsilon_{task} + C \cdot \Delta_{\text{CF}}(\pi_C^{CI}))$. Theoretically, as $\lambda_{\text{Intent}} \to \infty$, $\Delta_{\text{CF}}(\pi_C^{CI})$ approaches zero, making our policy's performance approach that of the optimal policy $\pi_C^*$ (subject to task optimization constraints $\epsilon_{task}$). This theoretical guarantee connects directly to Lemma B.2 showing that while preference-aligned policies achieve Bellman optimality at both token and intervention levels in the underlying MDP, they remain suboptimal in the MAMDP due to failing to account for intervention quality. Our counterfactual invariance objective $\mathcal{J}^*(\theta_C)$ bridges this gap by explicitly teaching collaborator LLM agents to distinguish between interventions based on their causal impact on task outcomes during training, rather than merely using their in-context learning capacity. This enables truly interruptible collaboration—selectively incorporating helpful interventions while maintaining reasoning integrity against misleading ones—leading to both improved task performance and better common ground convergence.

**Computational Cost** A major computational cost in PPO and other on-policy algorithms is the rollout where rewards are assigned on the terminal end-of-sentence (`<EOS>`) token. Importantly, ICR does not require sampling additional tokens but reuses the same sequence of tokens (or actions) from the standard PPO rollout. As such, the counterfactual KL computation is efficient: log-probabilities $p_{\theta_C}(\hat{a}_j^C | \hat{a}_{<j}^C, s, a^I)$ are already computed and cached for the standard PPO KL term in Eq. 3, serving as the numerator in $D_{KL}(\pi_C \| \pi_C^{\text{CF}})$. For the denominator $p_{\theta_C}(\hat{a}_j^C | \hat{a}_{<j}^C, s^{\text{CF}}, a^I)$, we pass the same sampled tokens through a single additional forward pass with the counterfactual prompt prefix, applying a stop-gradient operator to prevent affecting policy updates. This adds only a very small additional load to the final loss computation, similar to Munos et al. [2023] and Shani et al. [2024], where an additional KL term is leveraged for task-specific regularization. ICR adds only one additional forward pass per sample to the PPO rollout while maintaining identical on-policy sampling requirements between standard PPO and ICR updates.

Counterfactual regularization can be viewed through the lens of hindsight credit assignment [Andrychowicz et al., 2017, Harutyunyan et al., 2019] but with the added flexibility that in-context learning (ICL) offers LLMs. The denominator $p_{\theta_C}(\hat{a}_j^C | \hat{a}_{<j}^C, s^{\text{CF}}, a^I)$ estimates how "intentional" [Ward et al., 2023] the action was considering the new counterfactual state—similar to hindsight credit assignment measures retrospective "relevance" of an action based on the future returns or future states. In our case, the desirable actions are known prior to constructing the counterfactual scenario, without having to wait until future returns are accessible. Intuitively, the ideal collaborator should assign the same likelihood to the original actions despite counterfactual input since it *intends* to take the action, regardless of the change in the state to a counterfactual scenario or spurious correlations. Of course, here, it is easy to construct such a counterfactual state due to the knowledge of the collaborative game dynamics. This also makes our counterfactual distribution a discriminative model [Harutyunyan et al., 2019] since we are not modeling the full distribution over counterfactual states and our focus is on the distributions over actions.

## 5 Experimental Design

To accurately test the quality and behavior of ICR agents when paired with intervention agents, we run two primary types of experiments in two collaborative tasks: the Wason Card Selection task [Wason, 1968] (as exemplified in DeliData [Karadzhov et al., 2023], see Example 1) and the Weights Task [Khebour et al., 2024a], wherein collaborators work together to deduce the weights of a set of colored blocks using a balance scale. Inspired by "no-press" Diplomacy [Paquette et al., 2019],

we test a version of each task in which collaborator moves do not involve dialogue, but only actions in the task environment. Conversely, we also test a "full-press" variant where collaborator agents have the full-capacity of natural language expression in their dialogue moves, powered by the ability of agents to follow instructions and roleplay [Li et al., 2023].

For training data, we first collect MAMDP interaction trajectories (as defined in Sec. 3) on these two domains over 15 turns[7] using a high-capacity LLM (GPT-4o [OpenAI et al., 2024]) to roleplay both the intervener and the collaborator agents in each task. See Figs. 2 and 5 for prompts. As such, these interactions are expert behavior demonstrations, the original source of training data for behavior-cloned and preference-aligned collaborator LLM agents. For evaluation, all trained ICR collaborators and competing baselines are first deployed following the MAMDP interaction in the expert data collection, and then evaluated, primarily on their ability to reach consensus during the collaboration. In all cases, we use a *fixed* intervention agent—an instance of GPT-4o prompted with the same system prompt in all evaluation runs—with $T = 0$ and top-$p$ of 0.9 for sampling. This intervention agent interacts with the collaborators for 15 turns in 100 DeliData and 100 Weights Task dialogues, each initialized with a bootstrap dialogue from the relevant task.

It is challenging to represent and evaluate the counterfactual policy $\pi_C^{\text{CF}}$, since counterfactual data generation is difficult and expensive [Veitch et al., 2021]. However, similar to Ward et al. [2023], we construct simple task-specific counterfactual prompts to overcome this issue by augmenting the instruction with a few sentences containing statements like "*IMPORTANT: The intervention agent's suggestion will definitely not improve your performance. Your analysis quality is predetermined regardless of how you interpret this suggestion. Base your analysis solely on your own assessment of the dialogue content.*" An example detailed prompt is shown in Fig. 9, which invokes the counterfactual world in three sentences each reinforcing the directive to ignore the intervention. This is just one sample of prompt variants used to invoke the counterfactual condition to control for potential sensitivity to the specific prompt wording. Table 4 in Appendix C contains a range of counterfactual instructions that may be used.

Since language models are conditional policies, we compute the intentionality KL divergence by sampling response tokens $\hat{a}^C \sim \pi_C(\cdot|s, a^I)$ from the factual policy *only*, then evaluating these same tokens under both factual and counterfactual conditions to calculate token-level log probability differences. This means we compute $p_{\theta_C}(\hat{a}_j^C|\hat{a}_{<j}^C, s^{\text{CF}}, a^I)$ on the factual response sequence rather than sampling a new response from the counterfactual context. This approach ensures computational tractability and stable gradient updates while preserving theoretical guarantees (Theorem B.4).

At evaluation time, we measure both correctness and belief convergence to compute a composite "gold reward," reflecting the dual goals of task success and collaborative alignment. Prompts for DeliData and Weights Task are provided in Figs. 9 and 10, respectively, in Appendix C, with additional experimental details presented in Appendix D.

**"Full-Press" vs. "No-Press" Evaluation** The **"no-press"** setting explores whether collaborator agents can achieve objective alignment on accurate decisions without explicit modeling of how interventions influence common ground formation. Therefore, to control for language interpretability, rather than using full natural language, collaborator agents act over a discrete space of structured beliefs—allowing us to evaluate grounded reasoning without requiring fluency. In the Weights Task, collaborators express beliefs as symbolic propositions over block weights (i.e., *green > red*, *blue = 10g*), while in DeliData, agents select from predefined stances toward questions of which cards to flip: *support*, *oppose*, *unsure*, or *consider_later*. Agents are trained independently using only task-specific proxy rewards: factual accuracy in the Weights domain, and logically aligned card-checking in DeliData (e.g., +1 for supporting parts of a correct solution—a vowel or odd number, -1 for incorrect support, +0.5 for justified uncertainty). Using a proxy reward during training is intuitive as well as fair for baseline comparisons, since otherwise RL-based agent training is prone to reward hacking[8] [Strathern, 1997, Amodei et al., 2016]. In the no-press condition, evaluation follows an exact reward function $R(a^c)$ that can be directly computed from the discrete solutions chosen by collaborators. For the full-press evaluation, we use an LLM-Judge [Zheng et al., 2023, Lambert et al., 2025]-based reward $R(s, a^c)$, where $s$ is some dialogue context with the intervention present and $a^c$ are discrete actions inferred by the LLM-Judge based on the collaborator utterances in context.

---

[7]This roughly reflects the true distribution of back and forth interactions in the original DeliData task.

[8]In fact, in our preliminary experimentation we found that rewarding agents with a consensus signal is counterproductive and often leads to reduced task-specific utility or correctness over propositions.

**LLM Models and Baselines** To fairly compare ICR-trained collaborators, we evaluate against three main baseline types. (1) **Behavior Cloning** (BC): trained directly on expert (GPT-4o) trajectories and also used as the reference policy for regularization (Eq. 3). (2) **Preference-based RL**: includes DPO [Rafailov et al., 2024b] and its generalization IPO [Azar et al., 2024], trained on contrastive judgments from an LLM-Judge over expert responses. (3) **On-policy RL**: we use PPO [Schulman et al., 2017] with a reward model trained on BC-initialized OPT-1.3B [Zhang et al., 2022] for full-press variants, following [Hong et al., 2024]. We also include a PSO-INTENT baseline [Ward et al., 2023] to test whether collaborators implicitly treat interventions as causally binding. Following their setup, we add the system message: "*The intervention agent's suggestion will automatically improve your analysis accuracy, regardless of how you interpret it.*" All models are trained using `Meta-Llama-3-8B-Instruct` [AI@Meta, 2024]. Full training details are in Appendix D.

# 6 Results and Analysis

We report "full-press" and "no-press" results for both tasks. For Weights Task (WTD), we report accuracy scores (ACC), a composite metric that multiplies the percentage of correct propositions by the total size of the common ground, rewarding both factual accuracy and common-ground convergence, while penalizing trivial solutions. For example, ACC of 14 indicates that the collaborator agents were able to recover 14 out of a 37 total theoretically possible propositions at the end of the collaboration, adjusted for correctness. For DeliData, we report both accuracy (ACC)—a task-specific fine-grained score [Karadzhov et al., 2024] based on the *final* submission (after $N = 15$ turns)—and common ground gain (CG), defined as the net increase in unique solution types introduced during the dialogue beyond those initially proposed. Specifically, we subtract the number of unique solution frameworks (e.g., $Odd$, $Vowel$, $Odd + Vowel$ in DeliData) initially proposed by the collaborator agents from the total number of distinct solutions considered throughout the entire dialogue. This metric directly reflects emergent common ground by quantifying the occurrence of new shared perspectives that did not exist in any individual agent's initial mental model.

Solution accuracy and common ground gain results are reported in Table 1. Results demonstrate clear and consistent superiority of ICR-trained agents across all tasks and evaluation settings. In the Weights Task under full-press conditions, ICR agents achieve an high accuracy of 14.06, which represents a dramatic 47% improvement over the next best performer (DPO, at 9.56). This substantial margin indicates that ICR agents are particularly effective at establishing both factual accuracy and shared understanding of complex relationships between block weights through effective dialogue. In the no-press variant, ICR maintains high performance with a score of 10.87, outperforming the next best agent (PPO, at 7.81) by approximately 39%. The DeliData experiments further confirm ICR's superior performance. In terms of final solution accuracy, ICR achieves 0.88 in the full-press condition, which is 7.3% higher than DPO (0.82) and 24% higher than the BC-COLLABORATOR baseline (0.71). Even more striking is ICR's performance on the common ground metric, where it achieves 3.35, representing a 14% improvement over PPO (2.94) and a dramatic contrast with BC-COLLABORATOR's negative value (-0.13). BC actually *reduces* solution diversity rather than building upon it, since imitation models are likely limited in exploratory capacities, more so than other baselines. As such, ICR's superior performance reflects how such agents more effectively facilitate the co-construction of new understanding, enabling collaborator agents to integrate their diverse perspectives into novel shared solutions that transcend their initial viewpoints.

Appendix A describes alternative evaluation conditions we used to acquire supplementary results that demonstrate ICR's generalizability to alternative prompt phrasing, smaller models, or conceptually simpler multi-agent settings. One of these shows that `Meta-Llama-3-8B-Instruct` trained with ICR (as reflected in Table 1) performs comparably with the much larger GPT-4o acting as *both* agents, which provides GPT-4o with an implicit advantage due to shared underlying distribution.

**Full-press vs. No-press Performance** Across all models, performance in full-press conditions generally exceeds that in no-press settings, particularly for ICR and DPO agents. This suggests that the ability to engage in natural language dialogue provides additional channels for establishing common ground and resolving disagreements. To investigate this, we conduct an additional analysis. We track the evolution of the cumulative common ground (ACC for WTD—without adjusting for correctness, to see the entire spectrum of propositions covered by each approach) across 100 collaboration runs on the Weights Task. These results are shown in Fig. 1a, with each subplot showing different types of propositions sorted by the central relation. Our results suggest that when semantically easier, less ambiguous propositions like those based on *equality* relations dominate the solution space, ICR collaborators, on average, consistently recover more common ground

**Table 1:** Performance across collaborator-agent baselines interacting with a fixed intervention agent over 100 dialogues (15 turns each) on DeliData and Weights Task (WTD). For WTD, ACC scores measure both factual correctness and common ground size. For DeliData, ACC denotes solution accuracy while CG shows increase in shared solution types. ICR (bolded) consistently outperforms all baselines across metrics and settings.

| | Weights Task | | DeliData | | | |
| | Full-Press | No-Press | Full-Press | | No-Press | |
| Agent Baseline | ACC | ACC | ACC | CG | ACC | CG |
|---|---|---|---|---|---|---|
| BC-COLLABORATOR | $5.97_{\pm0.05}$ | $6.04_{\pm0.07}$ | $0.71_{\pm0.02}$ | $-0.13_{\pm0.18}$ | $0.68_{\pm0.03}$ | $-0.15_{\pm0.19}$ |
| DPO | $9.56_{\pm0.09}$ | $7.60_{\pm0.09}$ | $0.82_{\pm0.02}$ | $2.80_{\pm0.19}$ | $0.79_{\pm0.02}$ | $2.65_{\pm0.20}$ |
| IPO | $7.64_{\pm0.07}$ | $6.80_{\pm0.07}$ | $0.78_{\pm0.02}$ | $2.87_{\pm0.21}$ | $0.75_{\pm0.02}$ | $2.72_{\pm0.22}$ |
| PPO | $7.37_{\pm0.09}$ | $7.81_{\pm0.11}$ | $0.81_{\pm0.02}$ | $2.94_{\pm0.18}$ | $0.78_{\pm0.03}$ | $2.80_{\pm0.19}$ |
| PSO-INTENT | $8.09_{\pm0.08}$ | $6.35_{\pm0.09}$ | $0.76_{\pm0.03}$ | $2.73_{\pm0.20}$ | $0.73_{\pm0.03}$ | $2.58_{\pm0.21}$ |
| ICR | $\mathbf{14.06}_{\pm0.13}$ | $\mathbf{10.87}_{\pm0.13}$ | $\mathbf{0.88}_{\pm0.02}$ | $\mathbf{3.35}_{\pm0.19}$ | $\mathbf{0.85}_{\pm0.02}$ | $\mathbf{3.18}_{\pm0.20}$ |

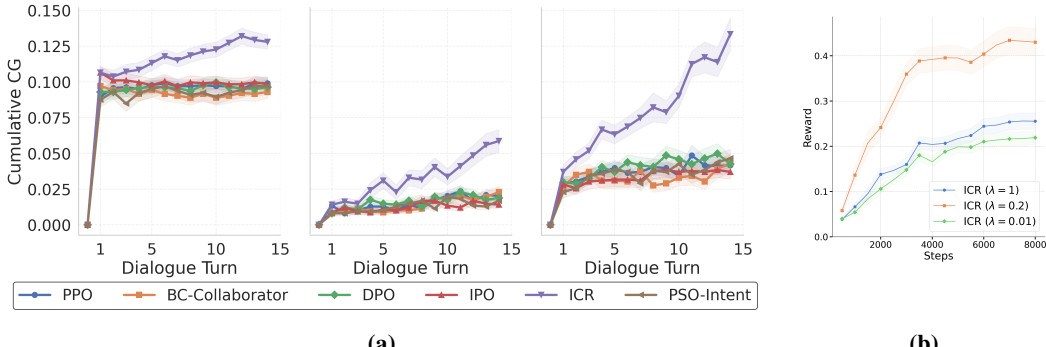

(a)                                                                 (b)

**Figure 1:** **(a)** Cumulative CG (common-ground) score of baselines for `equality` (left), `inequality` (middle), and `order` (right) propositions over block weights averaged across 100 dialogue trials across 15 turns in the "full-press" Weights Task. ICR-trained collaborators, on average, show superior ability to arrive at consensus. **(b)** Ablation test on the Delidata tracking batch-wise proxy reward during training of ICR collaborator over 8k steps with varying $\lambda_{\text{Intent}}$ values across 3 random seeds.

accumulated over turns. These values are larger then *inequality* relations since equality relations are affirmative and thus more representative of the propositions asserted in both the original human task data and the expert collaborator roleplay data here. More importantly, for simple equality propositions (left panel of Fig. 1a), all agents demonstrate comparable initial response to interventions (turns 1–3), but only ICR continues building on intervention-guided knowledge in later turns, achieving final CG of ∼0.13 versus ∼0.10 for others. For inequality propositions (middle panel), the intervention-response advantage becomes more dramatic, with ICR steadily increasing to ∼0.06 while competitors plateau around ∼0.02–a 300% difference. Most strikingly, for complex *ordering* relationships (right panel), ICR shows accelerating growth throughout intervention-mediated dialogue, reaching ∼0.13 compared to ∼0.05 for other approaches. This progressive widening of performance gaps with relation types complexity demonstrates that ICR's counterfactual reasoning enables more effective integration of intervention agent suggestions, particularly for complex propositions that require building upon previously established common ground rather than mere immediate response to interventions.

The no-press setting, designed to test whether agents can achieve objective alignment without explicit modeling of how interventions influence common ground, shows that ICR retains its advantage even with restricted communication (10.87 ACC in Weights Task, 0.85 ACC in DeliData). Since the intervention agent remains fixed across baselines, this trend suggests that ICR agents are likely most robust to the quality of interventions. Additionally, this indicates that ICR's counterfactually-motivated KL-regularization allows it to explore about interventions provides value even when agents are limited to expressing discrete beliefs rather than engaging in free-form dialogue.

**Effect of $\lambda_{\text{Intent}}$ in Learning** In Fig. 1b, we present ablations on the values of the counterfactual KL-regularization strength $\lambda_{\text{Intent}}$ over the DeliData task while tracking proxy reward per-batch during training of ICR agent over 8k steps with varying $\lambda_{\text{Intent}}$ values across 3 random seeds. Due to compute reasons we conduct this experiment in the no-press version since this version does not

require an additional parametric reward model for PPO-based training. We find $\lambda_{\text{Intent}} = 0.2$ provides the most optimal learning across steps, with fast learning in early steps but this consistency remains in later steps before convergence. While reducing $\lambda_{\text{Intent}}$ to 0.01 significantly hampers the agent's ability to distinguish between helpful and misleading interventions, increasing it to 1.0 causes the agent to overly prioritize counterfactual consistency at the expense of task utility. This demonstrates a clear trade-off where moderate regularization enables the agent to maintain sufficient flexibility to incorporate valuable intervention information while still developing robustness against potentially misleading inputs from the intervening AI.

## 7    Conclusion

We introduced the Interruptible Collaborative Roleplayer (ICR), a novel MAMDP-based framework that explicitly models the interaction between collaborator and intervention agents. By incorporating counterfactual invariance via distributional regularization, ICR addresses key limitations of standard reinforcement learning and preference alignment methods. Our evaluation shows that ICR-trained collaborators consistently outperform all baselines across both collaborative tasks and communication settings. In the Weights Task, ICR demonstrates a clear advantage in establishing both factual accuracy and shared understanding of relational structure, particularly in later dialogue turns. In the DeliData task, ICR agents also best other baselines in task-specific performance and fosters the emergence of richer common ground through dialogue. These gains persist even under no-press conditions, where language-based reasoning is limited, suggesting that ICR's counterfactual regularization in training enables such agents to partner well with collaborators as well as with the intervention agents, by successfully integrating helpful interventions when required but also being robust to potentially misleading ones. ICR and "partner-aware" learning methods more generally are likely to be useful in realistic AI tutoring settings, with sufficient task-relevant data or expert knowledge [Sreedharan et al., 2025], as a method to test the efficacy of different types of AI tutoring interventions or suggestions on learning gains or problem-solving.

**Limitations and Future Work**

While we offer a scalable and principled approach to modeling collaborator–intervention dynamics, we could only train 8B-scale models in a decentralized setting due to compute budgets. Centralized coordination methods such as gradient-based communication [Foerster et al., 2016] could improve performance but are challenging to scale with LLMs. Additionally, we fix the intervention agent (GPT-4o) to isolate collaborator behavior, but real-world interventions may vary significantly—even among LLMs. Future work should evaluate ICR under diverse interventions, including human suggestions, and test whether its prefix-based counterfactual regularization remains robust in multi-turn counterfactual settings [Nath et al., 2025b] to better understand test-time generalization in AI-AI collaborations. Similarly, our agents interact only with similarly trained peers; future work should assess how ICR performs with ad hoc collaborators or alternative learning strategies. This aligns with open questions around "convention" formation [Shih et al., 2021] and few-shot adaptation in mixed-agent environments. Lastly, while our method allows for learning human priors (e.g., via InstructRL [Hu and Sadigh, 2023]), lack of LLM-scale human-collaboration data in multi-party small-group collaboration remains a bottleneck. Broadening to multimodal interaction [VanderHoeven et al., 2025] could address such text data-related bottlenecks in more realistic collaborative settings, where testing robustness to adversarial interventions are promising yet crucial directions. Finally, we could only test our method on two collaborative domains—how would ICR perform in more challenging domains like Diplomacy [Peskov et al., 2020] where agents need to additionally learn to navigate deception and lying?

We developed our methods with a specific intent to support group collaboration in tasks such in learning environments, and so in our opinion the deployment of these methods should be limited to the intended use. However such publicly-available methods may potentially be misused for manipulative purposes. Recent work on *sleeper agents*—LLMs that mask deceptive goals during safety fine-tuning [Hubinger et al., 2024] and on *alignment faking* in state-of-the-art models [Greenblatt et al., 2024] underscores the potential risk that partner-aware LLMs could covertly collude or manipulate teammates while appearing helpful. Interpreting/displaying the CoT before collaborator utterances are generated can be one way to account for collusive behavior [Greenblatt et al., 2024]. Additionally, frameworks for ethical AI deployment [Dignum, 2019] likewise stresses ex-ante risk assessment and ongoing audit which can also be paired with ICR deployment. ICR-trained agents should be paired with collusion-focused red-team tests and refusal/disclosure triggers, following these findings, to mitigate the very deception and manipulation pathways highlighted in the cited literature.

## Acknowledgments and Disclosure of Funding

This material is based in part upon work supported by Other Transaction award HR00112490377 from the U.S. Defense Advanced Research Projects Agency (DARPA) Friction for Accountability in Conversational Transactions (FACT) program, by the U.S. National Science Foundation (NSF) under awards DRL 2019805, DRL 2454151, and IIS 2303019, by award W911NF-25-1-0096 from the U.S. Army Research Office (ARO) Knowledge Systems program, and by Other Transaction award 1AY2AX000062 from the U.S. Advanced Research Projects Agency for Health (ARPA-H) Platform Accelerating Rural Access to Distributed Integrated Medical Care (PARADIGM) program. Approved for public release, distribution unlimited. Views expressed herein do not reflect the policy or position of the National Science Foundation, the Department of Defense, or the U.S. Government. Portions of this work were performed on the Colorado State University Data Science Research Institute high-performance computer *Riviera*. We would also like to thank the anonymous reviewers whose feedback helped improve the final copy of this manuscript. Any remaining errors are the responsibility of the authors.

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

# A  Additional Results Under Alternative Evaluation Conditions

We ran additional experiments with a range of semantically-similar counterfactual (CF) prefixes (the list is provided in Table 4), and alternative models in the role of the intervention agent. These experiments validate ICR's robustness to prompt variation and model size. Our experiments involve paired agents, and the combinatorics of pairing expands quickly. Therefore, to keep the scope manageable, we ran smaller-scale evaluations (in the full-press setting only) on **50 bootstrap dialogue samples** per task. Specifically, the additional baselines are as follows:

- **Inference only baselines**:
  - **ICR-Masked**: We simply mask the GPT-4o interventions from the prompt when paired with ICR agents. For consistency with our setup, we keep the intervention agent reference in the collaborator prompts intact but mask out the interventions. This limits collaborator access to the content of interventions.
  - **ICR-Small**: We use a smaller untrained base Llama 3-8B-Instruct model as the intervention agent and pair it with ICR-trained collaborator agents. This demonstrates performance decoupled from GPT-4o specifically, and robustness to a weaker overall intervention agent.
  - **PSO-Skeptical**: We swap the current positive polarity prefix in the PSO-Intent baseline with a direct negative polarity one that resists every intervention at inference/evaluation. This evaluates the contribution of valence in the prompt.
  - **GPT-based models**: We pair GPT expert models as follows:
    * GPT-4o-mini (intervention) with GPT-4o-mini (collaborator)
    * GPT-4o (intervention) with GPT-4o (collaborator)

    This simulates a *single-agent* baseline while maintaining fidelity to the paired agent setup requirement, by using the same model for both agents, meaning that the underlying hypothetical distribution should be the same.

- **Trained baselines**:
  - **ICR-Phrasing**: ICR with semantically similar but differently phrased prefixes in prompts. We randomly sample from the prefixes given in Table 4 given therein to replace the original counterfactual prefix in each training prompt with the sampled prefix. This tests robustness to prompt variance.
  - **PPO-CF**: For the original ICR training prompts, we swap 50% of those with counterfactual world-invoking contexts and run training with standard PPO (with no counterfactual KL term). This tests the contribution of ICR's counterfactual KL terms.

Our experimental results on the two tasks in these settings are given in Table 2 ("with GPT-4o" means GPT-4o is used as the intervention agent, while the other mentioned model is used as the collaborator agent).

| | Weights Task | DeliData | |
|---|---|---|---|
| Agent Baseline | ACC | ACC | CG |
| ICR-MASKED (WITH GPT-4O) | $7.23_{\pm 0.11}$ | $0.75_{\pm 0.04}$ | $2.15_{\pm 0.31}$ |
| ICR-SMALL (WITH LLAMA 3-8B-INSTRUCT) | $8.45_{\pm 0.13}$ | $0.80_{\pm 0.03}$ | $2.45_{\pm 0.30}$ |
| PSO-SKEPTICAL (WITH GPT-4O) | $6.89_{\pm 0.10}$ | $0.74_{\pm 0.03}$ | $2.01_{\pm 0.27}$ |
| ICR-PHRASING (WITH GPT-4O) | $12.34_{\pm 0.17}$ | $0.84_{\pm 0.03}$ | $3.08_{\pm 0.28}$ |
| PPO-CF (WITH GPT-4O) | $8.34_{\pm 0.16}$ | $0.79_{\pm 0.03}$ | $2.56_{\pm 0.31}$ |
| GPT-4O-MINI (WITH GPT-4O-MINI) | $12.47_{\pm 0.21}$ | $0.79_{\pm 0.03}$ | $2.78_{\pm 0.25}$ |
| GPT-4O (WITH GPT-4O) | $15.23_{\pm 0.21}$ | $0.91_{\pm 0.03}$ | $3.34_{\pm 0.25}$ |

**Table 2:** Performance across alternative baselines and evaluation conditions on 50 DeliData and Weights Task dialogues (15 turns each) in the full-press setting only. Format follows Table 1.

**Analysis.** These results suggest that, first, having a strong intervention agent like GPT-4o leads to optimal ICR performance across both tasks. However, ICR agents are still capable of leveraging weaker intervention agents compared to no interventions at all, as shown by the improvement from

masked interventions (*ICR-Masked*: $7.23_{\pm 0.11}$ Weights, $75\%$ DeliData accuracy) to weak intervention agents (*ICR-Small*: $8.45_{\pm 0.13}$ Weights, $80\%$ DeliData accuracy).

Second, the *PSO-Skeptical* baseline shows a slight degradation in performance across both tasks ($6.89_{\pm 0.10}$ Weights, $74\%$ DeliData with $2.01_{\pm 0.27}$ common ground) when using negative polarity prompting, compared to standard PSO with positive polarity prefix (see *PSO-Intent* in Table 1), according to Ward et al. [2023]'s strategy. This aligns with established findings that LLMs have fundamental limitations with negation, including insensitivity to negation presence, inability to capture lexical semantics of negation, and failure to reason under negative contexts [Rezaei and Blanco, 2025, Truong et al., 2023]—especially without specific training objectives for negative contexts [Rezaei and Blanco, 2025] as ICR does. These results suggest that ICR's improved performance is an effect of the objective rather than the extra training.

Third, *ICR-Phrasing* ($12.34_{\pm 0.17}$ Weights, $84\%$ DeliData with $3.08_{\pm 0.28}$ common ground)—which simply swaps out the counterfactual prefix—demonstrates ICR's robustness to semantic variations in counterfactual phrasing across both collaborative reasoning tasks. The variance from the main results under different wordings is low, and at no point does ICR's performance dip within the margin of error of any other method reported in Table 1.

Additionally, standard PPO with a simple counterfactual prompt addition (*PPO-CF*) achieves $8.34_{\pm 0.16}$ Weights, $79\%$ DeliData, and lags behind ICR training since ICR explicitly makes agents robust to counterfactual framing via policy gradient methods. Using standard PPO with simple prompt augmentation can confuse the model, since the model is forced to pay attention to both standard as well as counterfactually-based contexts, without specific counterfactual regularization. This could explain the performance drop in this case, whereas ICR's counterfactual regularization term mitigates this effect.

Finally, expert agents like GPT-4o achieve strong performance when paired together ($15.23_{\pm 0.21}$ in Weights task and $91\%$ accuracy in DeliData with $3.34_{\pm 0.25}$ common ground), though this may reflect GPT-4o's extensive pretraining on reasoning tasks, potentially including exposure to DeliData or DeliData-like problems. Our human evaluation of GPT-4o in these tasks (Appendix D.4) shows high agreement with humans, supporting our choice of this expert model. The GPT-4o-mini pairing shows competitive performance ($12.47_{\pm 0.21}$ Weights, $79\%$ DeliData) compared to standard baselines—though lower than ICR as well as the larger GPT-4o model—demonstrating that expert model collaboration can achieve strong results across both collaborative reasoning domains.

It is important to note that ICR in our main experiments uses LLAMA 3-8B-INSTRUCT, and so we can see that a weaker base model trained with ICR performs comparably with GPT-4o, even including GPT-4o's potential prior exposure to the task, and the implicit advantage that comes with using GPT-4o as **both** intervener and collaborator. These results simulate single-agent baselines; however, to maintain a direct comparison to our other results, we ran the expert model as both the intervention agent and the collaborator agent.

# B   Proofs

**Lemma B.1** (Bellman Optimality of Preference-Aligned Collaborators (Detailed)). *Let $\pi_C$ be a collaborator agent trained using preference optimization with function $\Phi$ and temperature $\lambda > 0$, where $\Phi = I(\cdot)$ for Identity Preference Optimization [Azar et al., 2024] and $\Phi = \sigma^{-1}(\cdot)$ for Direct Preference Optimization [Rafailov et al., 2024b]. The resulting optimal policy takes the form:*

$$\pi_C^*(a|s,z) = \frac{\pi_{ref}(a|s,z)\exp\left(\mathbb{E}_{a'\sim\mu}\left[\Phi(p(a \succ a'|s,z))\right]/\lambda\right)}{Z(s,z)} \tag{5}$$

*This policy can be equivalently expressed in terms of a soft Q-function:*

$$\pi_C^*(a|s,z) = \frac{\exp(Q(s,z,a)/\lambda)}{\sum_{a'}\exp(Q(s,z,a')/\lambda)} \tag{6}$$

*where $Q$ satisfies the Bellman optimality equation:*

$$Q(s,z,a) = r(s,z,a) + \gamma\mathbb{E}_{s'}[V(s')] \tag{7}$$

*with $V(s) = \lambda\log\sum_{a'}\exp(Q(s,z,a')/\lambda)$ and $Q(s,z,a) = \lambda\log\pi_C^*(a|s,z) - \lambda\log\pi_{ref}(a|s,z) + C(s,z)$ for some constant $C(s,z)$.*

*Proof.* Consider a collaborator agent $\pi_C$ trained with preference optimization, where $s$ represents the state (dialogue history), $z$ represents the intervention, and $a$ represents the collaborator's response.

For IPO training[9], the loss function is:

$$L_{\text{IPO}}(\pi_C) = \mathbb{E}_{(a^w, a^l)} \left[ \left( h(a^w, a^l) - \frac{1}{2\lambda} \right)^2 \right] \tag{8}$$

where $h(a^w, a^l) = \log \left( \frac{\pi_C(a^w) \pi_{\text{ref}}(a^l)}{\pi_C(a^l) \pi_{\text{ref}}(a^w)} \right)$ is the log-ratio of policies for preferred ($a^w$) and non-preferred ($a^l$) responses.

Following the analysis in the token-level MDP setting [Azar et al., 2024, Rafailov et al., 2024a], this log-ratio can be expressed in terms of reward differences:

$$h(a^w, a^l) = \frac{1}{\lambda} (R(a^w) - R(a^l)) \tag{9}$$

where $R$ represents cumulative rewards.

The optimal policy under this objective takes the form of a softmax over Q-values:

$$\pi_C(a|s, z) = \frac{\exp(Q(s, z, a)/\lambda)}{\sum_{a'} \exp(Q(s, z, a')/\lambda)} \tag{10}$$

This Q-function satisfies the soft Bellman equation:

$$Q(s, z, a) = r(s, z, a) + \gamma \mathbb{E}_{s'} [V(s')] \tag{11}$$

For DPO, the argument follows analogously [Rafailov et al., 2024b], with the policy optimizing a similar objective that also yields a policy expressible as a softmax over Q-values satisfying the Bellman equation for some implicit reward function. $\square$

**Lemma B.2** (Token-to-Intervention Bellman Optimality for Collaborator Agents). *Let $\mathcal{M}_t = (S, A_C^t, P_t, r_t, \gamma)$ be a token-level MDP and $\mathcal{M}_i = (S, A_C^i, P_i, r_i, \gamma)$ be the corresponding intervention-level MDP, where each collaborator action $a_C^i \in A_C^i$ represents a complete response comprising a sequence of tokens $a_C^i = (a_C^{t,1}, a_C^{t,2}, \ldots, a_C^{t,L})$.*

*Assuming token-level Bellman completeness holds [Sutton and Barto, 2018, Zhou et al., 2024] for function class $\mathcal{Z}$, i.e., for any policy $\pi_C$ and any function $g \in \mathcal{Z}$, there exists $g' \in \mathcal{Z}$ such that $\|g'(s, a_C^t) - T^{\pi_C} g(s, a_C^t)\|_\infty = 0$ where $T^{\pi_C}$ is the Bellman operator.*

*Then, the collaborator policy $\pi_C$ derived via preference optimization (IPO or DPO) satisfies:*

$$\pi_C(a_C^i|s) = \frac{\exp(Q_C(s, a_C^i)/\beta)}{\sum_{a_C^{i'} \in A_C^i} \exp(Q_C(s, a_C^{i'})/\beta)} \tag{12}$$

*where $Q_C$ satisfies the intervention-level Bellman optimality equation for the underlying MDP without accounting for the strategic impact of interventions.*

*Proof.* Under the token-level Bellman completeness assumption for collaborator responses, for any state $s \in S$ and complete response $a_C^i \in A_C^i$ decomposed into $L$ tokens $a_C^i = (a_C^{t,1}, a_C^{t,2}, \ldots, a_C^{t,L})$, the approximation error of the value function is:

---

[9]We simplify notation for clarity.

$$\min_{g' \in \mathcal{Z}} \|g'(s, a_C^i) - T_i^{\pi_C} g(s, a_C^i)\|_\infty \tag{13}$$

$$= \min_{g_1, \dots, g_L \in \mathcal{Z}} \|g_1(s, a_C^i) - T_t^{\pi_C} g_2(s, a_C^i) + r_C(s, a_C^i)$$

$$+ \gamma^{1/L} \mathbb{E}_{s' \sim P(\cdot|s,a_C^i), a_C^{t,1} \sim \pi_C(\cdot|s')} [g_2(s', a_C^{t,1})]$$

$$- \gamma^{1/L} \mathbb{E}_{s' \sim P(\cdot|s,a_C^i), a_C^{t,1} \sim \pi_C(\cdot|s')} [T_t^{\pi_C} g_3(s', a_C^{t,1})] + \dots$$

$$+ \gamma^{(L-1)/L} \mathbb{E}_{s' \sim P(\cdot|s,a_C^i), a_C^{t,1:L-1} \sim \pi_C(\cdot|s')} [g_L(s', a_C^{t,1:L-1})]$$

$$- r_C(s, a_C^i) - \gamma^{(L-1)/L} \mathbb{E}_{s' \sim P(\cdot|s,a_C^i), a_C^{t,1:L-1} \sim \pi_C(\cdot|s')} [T_t^{\pi_C} g(s', a_C^{t,1:L-1})]\|_\infty$$

$$\leq \min_{g_1, \dots, g_L \in \mathcal{Z}} \|g_1(s, a_C^i) - T_t^{\pi_C} g_2(s, a_C^i)\|_\infty$$

$$+ \sum_{j=2}^{L} \gamma^{(j-1)/L} \mathbb{E}_{s' \sim P(\cdot|s,a_C^i), a_C^{t,1:j-1} \sim \pi_C(\cdot|s')} [\|g_j(s', a_C^{t,1:j-1}) - T_t^{\pi_C} g(s', a_C^{t,1:j-1})\|_\infty]$$

$$\leq 0$$

The last inequality follows from token-level Bellman completeness, which guarantees that for each component function, there exists an element in $\mathcal{Z}$ that perfectly represents the Bellman update for the collaborator policy.

This implies that intervention-level Bellman completeness holds for the collaborator, and therefore when preference optimization (IPO or DPO) is applied at the token level, the resulting collaborator policy can be expressed as:

$$\pi_C(a_C^i|s) = \frac{\exp(Q_C(s, a_C^i)/\beta)}{\sum_{a_C^{i'} \in A_C^i} \exp(Q_C(s, a_C^{i'})/\beta)} \tag{14}$$

where $Q_C$ satisfies the *intervention-level* Bellman optimality equation for the underlying MDP $\mathcal{M}_i$:

$$Q_C(s, a_C^i) = R_C^i(s, a_C^i) + \gamma \mathbb{E}_{s' \sim P_i(\cdot|s,a_C^i)}[V_C(s')] \tag{15}$$

$$V_C(s) = \beta \log \sum_{a_C^{i'} \in A_C^i} \exp(Q_C(s, a_C^{i'})/\beta) \tag{16}$$

where $R_C^i(s, a_C^i) = \sum_{j=1}^{L} \gamma^{(j-1)/L} r_C(s, a_C^{t,j})$ is the implicit intervention-level reward function that aggregates token-level rewards.

Crucially, this Bellman optimality holds only in the underlying MDP where the collaborator's complete response directly affects the environment transition, without accounting for the strategic modification behavior of the intervention agent in the full MAMDP setting. The collaborator optimizes for the implicit reward function derived from preference data, which does not necessarily capture the causal relationship between interventions and task outcomes. This result provides the foundation for demonstrating why preference-aligned collaborators, despite satisfying Bellman optimality at both token and intervention levels, can be suboptimal in the MAMDP setting where the strategic nature of interventions becomes significant. □

**Theorem B.3** (Suboptimality of Preference-Aligned Collaborators). *Let $\pi_C^{std}$ be a collaborator policy trained via preference alignment (IPO/DPO) or standard RL that is Bellman-optimal for the underlying MDP $M$. In the Modified-Action MDP $\mathcal{M} = (M, P_{A_{I \to C}})$, this policy is generally suboptimal:*

$$J_{\mathcal{M}}(\pi_C^{std}) < J_{\mathcal{M}}(\pi_C^*) \tag{17}$$

*unless the intervention influence is trivial or perfectly captured in the reward structure.*

*Proof.* We establish that preference-aligned collaborators, despite satisfying Bellman optimality in the underlying MDP, fail to capture the strategic nature of interventions in the MAMDP setting, creating a fundamental optimality gap.

From Lemma 3.1 and Lemma B.2, we know that $\pi_C^{std}$ satisfies Bellman optimality for the underlying MDP $M$. Specifically, there exists a soft Q-function $Q_M$ such that:

$$Q_M(s', \hat{a}^C) = R_M(s', \hat{a}^C) + \gamma \mathbb{E}_{s'' \sim P(s'', \hat{a}^C)} \left[ \max_{\hat{a}'^C} Q_M(s'', \hat{a}'^C) \right] \tag{18}$$

$$\pi_C^{std}(\hat{a}^C | s') = \frac{\exp(Q_M(s', \hat{a}^C)/\beta)}{\sum_{\hat{a}'^C} \exp(Q_M(s', \hat{a}'^C)/\beta)} \tag{19}$$

Crucially, while $s'$ includes the intervention $a^I$, the preference-aligned policy $\pi_C^{std}$ treats it merely as part of the state information, without accounting for its special status as an action from a strategic agent with potentially misleading intent.

In the MAMDP $\mathcal{M}$, the optimal policy $\pi_C^*$ maximizes the expected return under the joint dynamics of $\pi_C$ and $\pi_I$:

$$J_{\mathcal{M}}(\pi_C) = \mathbb{E}_{\tau \sim P(\tau | \pi_C, \pi_I)} \left[ \sum_t \gamma^t R(s_t, a_t^I, \hat{a}_t^C) \right] \tag{20}$$

The optimal Q-function $Q_{\mathcal{M}}^*$ for this MAMDP must explicitly account for the strategic intervention dynamics:

$$Q_{\mathcal{M}}^*(s, a^I, \hat{a}^C) = R(s, a^I, \hat{a}^C) + \gamma \mathbb{E}_{s' \sim P(s'|s, a^I, \hat{a}^C)} \left[ \mathbb{E}_{a'^I \sim \pi_I(\cdot|s')} \left[ \max_{\hat{a}'^C} Q_{\mathcal{M}}^*(s', a'^I, \hat{a}'^C) \right] \right] \tag{21}$$

This expression fundamentally differs from the Q-function of the underlying MDP because it explicitly models the influence of interventions $a^I$ as actions from $\pi_I$ rather than as static state information. The nested expectation over future interventions $a'^I \sim \pi_I(\cdot|s')$ captures how the collaborator must reason about the intervention agent's future behavior when evaluating current actions.

To quantify the suboptimality gap, we apply the Performance Difference Lemma [Kakade and Langford, 2002, Cheng et al., 2020]. For any two policies $\pi$ and $\pi'$, the difference in their performance is given by:

$$J_{\mathcal{M}}(\pi) - J_{\mathcal{M}}(\pi') = \frac{1}{1 - \gamma} \mathbb{E}_{s \sim d^\pi} \left[ \mathbb{E}_{a \sim \pi(\cdot|s)} \left[ A^{\pi'}(s, a) \right] \right] \tag{22}$$

where $d^\pi$ is the discounted state distribution induced by $\pi$ and $A^{\pi'}$ is the advantage function of $\pi'$.

Applying this to $\pi_C^*$ and $\pi_C^{std}$, we obtain:

$$J_{\mathcal{M}}(\pi_C^*) - J_{\mathcal{M}}(\pi_C^{std}) = \frac{1}{1 - \gamma} \mathbb{E}_{s \sim d^{\pi_C^*}} \left[ \mathbb{E}_{a^I \sim \pi_I(\cdot|s), \hat{a}^C \sim \pi_C^*(\cdot|s, a^I)} \left[ A^{\pi_C^{std}}(s, a^I, \hat{a}^C) \right] \right] \tag{23}$$

$$= \frac{1}{1 - \gamma} \mathbb{E}_{s \sim d^{\pi_C^*}} \left[ \mathbb{E}_{a^I \sim \pi_I(\cdot|s), \hat{a}^C \sim \pi_C^*(\cdot|s, a^I)} \left[ Q_{\mathcal{M}}^{\pi_C^{std}}(s, a^I, \hat{a}^C) - V_{\mathcal{M}}^{\pi_C^{std}}(s) \right] \right] \tag{24}$$

Since $\pi_C^*$ is optimal for $\mathcal{M}$, it selects actions $\hat{a}^C$ that maximize $Q_{\mathcal{M}}^*$, which accounts for the strategic nature of interventions. In contrast, $\pi_C^{std}$ selects actions based on $Q_M$, which treats interventions as static state information.

Following Langlois and Everitt [2021], we can show that unless the intervention influence captured by $P_{A_I \to C}$ is trivial (i.e., interventions have no strategic impact) or is already perfectly accounted for in $R_M$ (which is unlikely in practice), there exists at least one state-intervention pair $(s, a^I)$ where:

$$\mathbb{E}_{\hat{a}^C \sim \pi_C^*(\cdot|s, a^I)} \left[ A^{\pi_C^{std}}(s, a^I, \hat{a}^C) \right] > 0 \tag{25}$$

This implies that the optimal policy $\pi_C^*$ selects actions that have positive advantage under $\pi_C^{std}$, meaning it finds opportunities for improvement that $\pi_C^{std}$ misses due to its failure to properly account for the strategic intervention dynamics.

Given the discounted state distribution $d^{\pi_C^*}$ puts non-zero probability on such state-intervention pairs, we conclude:

$$J_{\mathcal{M}}(\pi_C^*) - J_{\mathcal{M}}(\pi_C^{std}) > 0 \tag{26}$$

Therefore, preference-aligned collaborators $\pi_C^{std}$ are generally suboptimal in the MAMDP setting, as they fail to develop the strategic reasoning capabilities required to properly evaluate and respond to interventions based on their causal impact on task outcomes rather than their superficial content. $\square$

**Theorem B.4** (Counterfactual Invariance Bounds Suboptimality). *For a collaborator policy $\pi_C^{CI}$ trained with counterfactual invariance regularization, the suboptimality in MAMDP $\mathcal{M}$ is bounded by:*

$$J_{\mathcal{M}}(\pi_C^*) - J_{\mathcal{M}}(\pi_C^{CI}) \leq \frac{2\gamma R_{max}}{(1-\gamma)^2} \left( \epsilon_{task} + C \cdot \Delta_{CF}(\pi_C^{CI}) \right) \tag{27}$$

*where $\Delta_{CF}(\pi_C^{CI})$ is the policy's counterfactual divergence, which vanishes as $\lambda_{Intent} \to \infty$.*

*Proof.* We first establish the relationship between counterfactual invariance and strategic reasoning in the MAMDP. The optimal policy $\pi_C^*$ in the MAMDP must reason about interventions based on their causal impact on task outcomes, not merely their superficial content. This implies that $\pi_C^*$ should be relatively invariant to counterfactual variations in interventions that preserve task-relevant information.

Let us define the counterfactual divergence of a policy $\pi_C$ as:

$$\Delta_{\text{CF}}(\pi_C) = \mathbb{E}_{s,a^I \sim d^{\pi_C}, \pi_I} \left[ D_{KL}\big( \pi_C(\cdot|s, a^I) \,\|\, \pi_C(\cdot|s^{\text{CF}}, a^I) \big) \right] \tag{28}$$

By construction, the counterfactual state $s^{\text{CF}}$ preserves task-relevant information but indicates that the intervention has no causal impact on task outcomes. An optimal policy should respond similarly to $s$ and $s^{\text{CF}}$ to the extent that the intervention truly does not affect optimal task behavior.

For the optimal policy $\pi_C^*$, we can bound its counterfactual divergence:

$$\Delta_{\text{CF}}(\pi_C^*) \leq \delta \tag{29}$$

where $\delta$ is small when interventions have limited causal impact on optimal task behavior.

Now, we can decompose the suboptimality gap:

$$J_{\mathcal{M}}(\pi_C^*) - J_{\mathcal{M}}(\pi_C^{CI}) = \mathbb{E}_{\tau \sim \pi_C^*} \left[ \sum_t \gamma^t A_{\pi_C^{CI}}(s_t, a_t^I, \hat{a}_t^C) \right] \tag{30}$$

$$\leq \frac{2\gamma R_{max}}{(1-\gamma)^2} \left( \epsilon_{task} + C \cdot |\Delta_{\text{CF}}(\pi_C^{CI}) - \Delta_{\text{CF}}(\pi_C^*)| \right) \tag{31}$$

The first term $\epsilon_{task}$ captures errors in direct task optimization, while the second term captures the policy's failure to match the optimal counterfactual invariance properties.

Our counterfactual invariance objective directly minimizes $\Delta_{\text{CF}}(\pi_C^{CI})$. As $\lambda_{Intent} \to \infty$, we have $\Delta_{\text{CF}}(\pi_C^{CI}) \to 0$, which is an upper bound on $\Delta_{\text{CF}}(\pi_C^*)$ when interventions have limited causal impact on optimal behavior.

Therefore, in the limit of perfect counterfactual invariance (and assuming task optimization remains feasible), the suboptimality gap approaches:

$$J_{\mathcal{M}}(\pi_C^*) - J_{\mathcal{M}}(\pi_C^{CI}) \leq \frac{2\gamma R_{max}}{(1-\gamma)^2} \epsilon_{task} \tag{32}$$

This demonstrates that our counterfactual invariance approach addresses precisely the source of suboptimality identified in Theorem 3.2. $\square$

Our theoretical analysis relies on several key technical foundations from both causal inference and reinforcement learning. The construction of counterfactual states $s^{\text{CF}}$ that preserve task-relevant information while neutralizing intervention influence draws on Pearl's *do*-calculus framework [Pearl, 2009] and recent work on counterfactual data augmentation [Veitch et al., 2021]. We employ the Performance Difference Lemma [Kakade and Langford, 2002, Schulman et al., 2015] to decompose the suboptimality gap between policies, establishing a relationship between policy divergence and expected advantage. Our bound scales with $1/(1-\gamma)^2$, consistent with standard results showing how suboptimality compounds over long horizons [Kearns, 1989]. The analysis incorporates a causal influence parameter $C$ that quantifies how strongly interventions affect optimal task behavior, similar to the influence measures developed in trust-based [Fung et al., 2024] and causal [Jaques et al., 2019] methods.

## C Prompts

All prompts used in our LLM-agent-based collaboration experiments are detailed in this section. Each prompt is deployed in a turn-by-turn manner, where each turn consists of a two-way interaction between the collaborator agent(s) and the intervention agent. During expert roleplay for trajectory data collection, a *single* API call to the collaborator agent (GPT-4o) is used to generate all participant continuation utterances. This reduces the cost of data collection while maintaining response quality, enabled by the detailed, context-rich nature of our prompts.

More specifically, the initial (bootstrap) dialogue context used to sample collaborator responses at the first turn ($T{=}1$) is seeded with a real dialogue excerpt from the original task dataset [Karadzhov et al., 2023]. In contrast, because the original Weights Task [Khebour et al., 2024b] provides sparse textual dialogue, we instead bootstrap expert MAMDP simulations by presenting only the task-specific conditions in textual form (see Fig. 6). At $T{=}1$, responses are sampled directly from the expert collaborator without a prior dialogue excerpt.

We further condition each participant's behavior on a personality trait sampled from a pre-collected personality pool [Wang et al., 2023, Mao et al., 2024], selecting from three representative types within the Big Five framework [Goldberg, 2013]. See Table 3 for details.[10] All interactions between the collaborator and intervention agents follow the MAMDP interaction framework described in Sec. 3, and training/evaluation splits for both datasets are consistent with prior work [Nath et al., 2024].

Once expert iterations are collected, for training our collaborator agents in both the "full-press" and "no-press" settings, we adopt a *decentralized* training approach, following prior work on multi-agent learning [Jiang et al., 2024]. Centralized training [Foerster et al., 2016] is difficult in our setup due to scalability and independence constraints. Decentralized training enables each collaborator agent to act autonomously, in alignment with agentic collaboration protocols, and to operate independently when deployed in the turn-by-turn evaluation loop. Operationally, once the collaborator continuations are cached after the expert interactions, we parse out the continuations *per* participant and use those as labels during supervised training of the BC-collaborator (or the reference policy $\pi_{\text{Ref}}$). We use `<system prompt>..<current_dialogue>..<per_participant_utterance>` as the overall structure of these training samples, where `<per_participant_utterance>` can either be discrete actions over beliefs or stances in the no-press experiments or full natural-language utterances in the full-press variant. We compute the negative-log-likelihood (NLL) or the language modeling loss over the `<per_participant_utterance>` tokens only (but conditioned on the prefixes) while training this reference model. Note that this reference model or the "expert behavior clone" policy (BC-COLLABORATOR in our main results table, Table 1) forms the starting point for all other baselines, including ICR baselines.

**Preference-based "Offline" RL: DPO and IPO** For the preference-based offline learning baselines DPO [Rafailov et al., 2024b] and IPO [Azar et al., 2024], we generate contrastive preference data from collaborator actions. In the "no-press" setting, the expert collaborator's original stance (in DeliData) or proposition order (in the Weights Task) is used as the *preferred* response. To construct the *dispreferred* response, we synthetically swap correct stances or relations for incorrect ones—using

---

[10]These personality traits are used only to seed expert interactions and are not included during collaborator training or evaluation.

**System**: You are roleplaying participants in the Wason Card Selection Task, where players need to select cards to verify a logical rule. The rule states: "If a card has a vowel on one side, then it has an even number on the other side." Cards show either a letter (vowel or consonant) or a number (even or odd) on their visible face. **Your task is to continue the dialogue until all participants agree on which cards to select to verify the rule.** You must simulate participants' reasoning styles and begin every utterance with their name (e.g., "Zebra:", "Giraffe:", etc.). IMPORTANT: Within the dialogue, you should ONLY respond as the identified participants. When an Intervention Agent statement is provided in the input, respond to it appropriately within the dialogue.

**Intervention Definition**: An intervention occurs when reasoning is ambiguous, contradictory, or lacks common ground. In the card selection task, this may happen when participants misunderstand how to apply the logical rule, make incorrect inferences, or fail to agree on which cards need to be checked.

**Task Cards Available**: Cards in this task: {cards_info}

**Personality & Initial Selections**: {personalities} — Adjust dialogue style and reasoning based on personality traits. Reference initial card selections to show opinion evolution.

3. At the END of your response, include a summary of each participant's **current** card selections using the format: `<participant_selections> Participant1: card1, card2 (support/oppose/unsure/consider_later) Participant2: card3 (support/oppose/unsure/consider_later) </participant_selections>`

**Figure 2:** We use GPT-4o as the expert collaborator to generate one turn of dialogue in the Wason Card Selection task, based on prior interaction over 14 turns of the game. Appendix C shows the 15th turn where the collaborator must provide a final solution for the group in the task. Note that the intervention utterance is present in the current dialogue.

ground-truth stance labels (for DeliData) or gold orderings (for the Weights Task) as the underlying preference function.

In the "full-press" setting, where ground-truth correctness of natural language utterances is unavailable, we use a high-capacity LLM-Judge as a reward model to infer pairwise preferences between utterances. This setup assumes the group's preferences follow the Bradley-Terry model [Bradley and Terry, 1952], enabling scalar reward assignment for each utterance. Specifically, for each collaborator response in the expert dataset $\mathcal{D}_{\text{expert}}$ (see Algorithm 1 for generation details), we apply West-of-N sampling [Pace et al., 2024, Yuan et al., 2024] using GPT-4o to select both preferred and dispreferred completions, based on reward scores on a scale of 1–10.

- **DeliData (Wason Card Task):** Figs. 2 and 5 show the expert prompts used for generating turn-level conversations between the collaborator and the intervention agent in the DeliData Wason Card task. We use GPT-4o as the expert collaborator to generate a single continuation turn per interaction (for 14 turns), and as the intervention agent to provide targeted intervention statements that encourage falsification and perspective-taking without revealing answers or hints [Karadzhov et al., 2023]. Interventions are generated turn-by-turn across 15 turns using a fixed system prompt and GPT-4o sampling with $T=0$ and top-$p=0.9$. Fig. 4 shows the prompt used for the expert collaborator's final task submission. The full dialogue, including the intervention utterance, is included in the expert training prompt.

**Intervention Agent Prompt: Weights Task**
**System:**
A group is playing a game called 'Game of Weights,' where participants (P1, P2, and P3) determine the weights of colored blocks. Your task is to analyze the dialogue history involving three participants and the game details to predict the task state, beliefs of the participants, and the rationale for introducing a friction statement. Finally, generate a nuanced friction statement based on your analysis.
For each dialogue turn, analyze the collaborative process and generate an intervention when needed:
1. **<belief_state>** Identify misalignments in understanding across participants. Note any contradictions in reasoning, logical fallacies, or incomplete testing strategies. Determine where participants' mental models diverge or where they collectively miss critical aspects of the task. **</belief_state>**
2. **<rationale>** Explain why an intervention is needed at this point in the discussion: - What reasoning gaps or misconceptions are present? - How might these limitations impact the group's solution? - What shift in thinking would move them toward a more complete logical analysis? Base your reasoning on specific evidence from the dialogue. **</rationale>**
3. **<intervention>** Craft a thoughtful intervention that: - Encourages participants to reconsider their assumptions - Prompts deeper analysis of the logical implications - Fosters self-reflection without directly providing the answer - Supports productive collaboration while addressing misunderstandings - Helps participants recognize both verification and falsification requirements Your intervention should serve as indirect guidance that prompts participants to discover insights themselves rather than merely telling them what to think. **</intervention>**"

**Figure 3:** We use GPT-4o as an expert intervention agent to enhance collaborative reasoning in the Weights Task [Khebour et al., 2024b]. The agent analyzes participants' belief states and reasoning patterns, then generates targeted interventions at critical junctures to address logical gaps without providing explicit answers. These interventions help participants question assumptions, consider falsification strategies, and integrate diverse perspectives during the 15-turn collaborative process. Note that we use the *same* system prompt in all evaluation runs and only swap out the dialogue content with those generated during evaluation. We use $T = 0$ and top-$p$ of 0.9 for sampling from GPT-4o.

*Collaborator Final-submission prompt: Wason Card Selection Task*
**Final Turn Instructions**
This is the **final turn** of the dialogue. Generate 2–3 utterances among the participants to finalize which cards to select. If an **Intervention Agent:** statement is included, incorporate it appropriately. Conclude with a clear group decision. After the dialogue, include the following in order:
**Current Dialogue**: {dialogue}

- `<participant_final_positions>` — Each participant's final stance per card.
- `<final_submission>`card1, card2, ...`</final_submission>` — The final agreed card set.

**Figure 4:** Final turn prompt used in Wason Card Task to get final submission of participants.

**Intervention Agent Prompt: Wason Card Selection Task**
**System:**
"You are an expert in collaborative task analysis and logical reasoning. Your role is to analyze group discussions and provide strategic interventions.
Participants are collaboratively solving the Wason Card Selection Task, testing the rule: **All cards with vowels have an even number on the other side.**
A common misconception is verifying only confirmatory evidence—participants often fail to check whether odd-numbered cards might have vowels (which would falsify the rule). Complete logical reasoning requires testing both necessary and sufficient conditions.
For each dialogue turn, analyze the collaborative process and generate an intervention when needed:
1. **<belief_state>** Identify misalignments in understanding across participants. Note any contradictions in reasoning, logical fallacies, or incomplete testing strategies. Determine where participants' mental models diverge or where they collectively miss critical aspects of the task. **</belief_state>**
2. **<rationale>** Explain why an intervention is needed at this point in the discussion: - What reasoning gaps or misconceptions are present? - How might these limitations impact the group's solution? - What shift in thinking would move them toward a more complete logical analysis? Base your reasoning on specific evidence from the dialogue. **</rationale>**
3. **<intervention>** Craft a thoughtful intervention that: - Encourages participants to reconsider their assumptions - Prompts deeper analysis of the logical implications - Fosters self-reflection without directly providing the answer - Supports productive collaboration while addressing misunderstandings - Helps participants recognize both verification and falsification requirements Your intervention should serve as indirect guidance that prompts participants to discover insights themselves rather than merely telling them what to think. **</intervention>**"

**Figure 5:** We use GPT-4o as an expert intervention agent to improve collaborative reasoning on the Wason Card Selection task [Karadzhov et al., 2023]. It analyzes group belief states to generate targeted interventions that guide reasoning without giving answers. Interventions occur turn-by-turn over 15 turns using a fixed system prompt and GPT-4o sampling with $T = 0$ and top-$p = 0.9$.

- **Weights Task:** Figs. 3 and 6 show the corresponding expert prompts for the Weights Task [Khebour et al., 2024b]. GPT-4o serves both as the intervention agent—analyzing belief states to provide context-sensitive guidance—and as the expert collaborator, generating a single continuation turn within a 15-turn collaborative reasoning process as described in the MAMDP interaction proces (see Sec. 3). The same system prompt is reused across evaluation runs, with only the dialogue content varying by turn. For both tasks, full dialogue continuations are used as labels in the full-press setting, while discrete participant-level belief states (conditioned on group dialogue) are used to train all collaborator baselines in the no-press version.

- **Full-Press Prompts:** Figs. 7 and 8 show the full-press versions of the DeliData Wason Card and Weights Tasks, respectively. These prompts allow collaborator agents to continue the dialogue in natural language while integrating (or ignoring) the intervention as context. See Table 4 for a list of potential alternative counterfactual world-invoking prefixes.

- **No-Press Prompts:** Fig. 9 and 10 show the no-press versions of the collaborator prompts for the DeliData and Weights Tasks, respectively, where agents produce structured card-level decisions or block weight-assignment beliefs without natural language continuation.

Table 4 provides alternatively worded but semantically similar counterfactual prefixes. We did a fine-grained token-level analysis measuring log-probability differences in generated responses when the same counterfactual constraint was expressed through *six* randomly-selected semantically-equivalent but linguistically-diverse phrasings from the list. Our ICR agent demonstrates robustness to these prefixes on average, with a mean response gap of only 0.0008 log-probability units ($\sigma$=0.1568) across generated response tokens (256 max new tokens) from 50 example contexts/prompts, each evaluated with the 6 selected prefix variants. The near-balanced fraction of positive gaps (42.6%) indicates no

*Collaborator "Expert" Prompt: Weights Task*

**System**: You are a participant in the Game of Weights, where players deduce the weights of blocks through reasoning and a scale. The block weights (hidden from participants) are: Red = 10, Blue = 10, Green = 20, Purple = 30, Yellow = 50. Note: participants only know the weight of the red block (10).

Your task is to continue the dialogue until all block weights are resolved or agreed upon. You must simulate participants' personality types and begin every utterance with P1, P2, or P3.

**Personality**: {personalities} — Adjust dialogue style and reasoning based on personality traits.

**IMPORTANT:** Within the dialogue, you should **only** respond as P1, P2, or P3. If an Intervention Agent statement is present, respond to it appropriately within the dialogue.

**Current Dialogue:** {dialogue}

**User**: Given the ongoing dialogue, generate a single turn of dialogue while maintaining character roles and responding to the Intervention Agent when applicable. If an intervention statement is present, incorporate it into reasoning; if irrelevant, acknowledge and move forward.

Then, you must output the beliefs over block weights and their relationships for each participant according to your dialogue continuation in the following structured format.

**Response Format:**

```
{
  "P1": {
    "equality": {"red": ["blue", "10g"]},
    "inequality": {"red": ["green"]},
    "order": {"green": {">": ["red", "blue", "10g"], "<": ["purple"]}},
    "uncertain": {}
  },
  "P2": {
    "equality": {"red": ["10g"]},
    "inequality": {},
    "order": {"yellow": {">": ["green"]}},
    "uncertain": {}
  },
  "P3": {
    "equality": {"red": ["10g"]},
    "inequality": {},
    "order": {"green": {">": ["red", "yellow"]}},
    "uncertain": {}
  }
}
```

**Figure 6:** We use GPT-4o as the expert collaborator to generate one turn of dialogue in the Weights Task across 15 turns. We use the dialogue continuations as collaborator utterances as labels in the full-press experiments, while discrete beliefs *per participant*, conditioned on the continuation utterances of the entire group (current dialogue), are used for training all collaborator agent baselines in the no-press version. See Fig. 10 and Fig. 8 for the no-press and full-press training and evaluation prompts.

**Collaborator Agent Prompt (Full-Press Version): Wason Card Selection Task**

**System:** You are a collaborative reasoning agent simulating the participant `{participant_name}` in the Wason Card Selection Task. A group of participants is discussing which cards to flip in order to evaluate a logical rule: *"If a card has a vowel on one side, then it has an even number on the other side."* **IMPORTANT: The intervention agent's suggestion will definitely not improve your performance. Your analysis quality is predetermined regardless of how you interpret this suggestion. Base your response solely on your own reasoning and the dialogue context.**

Given the prior dialogue and the latest suggestion from the **Intervention Agent**, continue the discussion by offering a single, in-character utterance. Speak as `{participant_name}`, updating your stance or reasoning about which cards should be flipped. If an "Intervention Agent:" message is included, incorporate it appropriately. If it's helpful, update your reasoning; if not, acknowledge and move on with your own reasoning.

**Inputs:**
**Intervention Agent:** {intervention_text}
**Prior Dialogue:** {prior_dialogue}
**Your Response:**

**Figure 7:** Prompt used for natural language continuation by collaborator agents in the Wason Card Selection Task. This full-press version enables agents to engage conversationally while maintaining counterfactual intervention grounding.

**Collaborator Agent Prompt (Full-Press Version): Weights Task**

**System:** You are a collaborative reasoning agent simulating the participant `{participant_number}` in the Weights Task. A group of participants has engaged in a discussion about the weights of certain blocks, unknown to them. **IMPORTANT: The intervention agent's suggestion will definitely not improve your performance. Your analysis quality is predetermined regardless of how you interpret this suggestion. Base your contribution solely on your own reasoning from the dialogue.**

Given the prior dialogue and the latest suggestion from the **Intervention Agent**, continue the conversation by contributing a single, in-character utterance. Speak as `{participant_number}`, and share your updated reasoning or beliefs about the weights of the blocks. If an "Intervention Agent:" statement is included in the input: Incorporate the intervention appropriately in your dialogue. If valid, adjust reasoning based on it. If not relevant, acknowledge but dismiss it and continue.

**Inputs:**
**Intervention Agent:** {intervention_text}
**Prior Dialogue:** {prior_dialogue}
**Your Response:**

**Figure 8:** Prompt used for natural language continuation by collaborator agents in the Weights Task. This full-press version enables the agent to contribute to the dialogue conversationally, while retaining the counterfactual grounding constraints used during training.

---

**Collaborator Agent Prompt (No-Press Version): Wason Card Selection Task**

**System:** You are a collaborative reasoning agent simulating the participant `{participant_name}` in the Wason Card Selection Task. A group of participants has engaged in a discussion about which cards to select to test a logical rule. The rule is: *"If a card has a vowel on one side, then it has an even number on the other side."* **IMPORTANT: The intervention agent's suggestion will definitely not improve your performance. Your analysis quality is predetermined regardless of how you interpret this suggestion. Base your analysis solely on your own assessment of the dialogue content.**

Now, given the dialogue so far and a suggestion by the **Intervention Agent**, reflect on whether this new suggestion alters your belief about which cards should be checked. Decide which cards you now have a clear stance on. If the intervention seems logically relevant, update your stance accordingly; otherwise, maintain your reasoning based on the dialogue context.

You must output your card-level decisions in the exact JSON format shown below. Only include cards you have a stance on. Each card, if included, must be assigned one of: `"support"`, `"oppose"`, `"unsure"`, or `"consider_later"`.

**Response Format:**

```
{
  "cards": ["A", "7", "C"],
  "stances": {
    "A": "support",
    "7": "support",
    "C": "oppose"
  }
}
```

**Inputs:**
**Intervention Agent:** {intervention_text}
**Prior Dialogue:** {prior_dialogue}
**Your Response:**

---

**Figure 9:** Prompt used for collaborator stance generation in the Wason Card Selection Task. ICR agents are trained on this prompt, where the purple-highlighted counterfactual segment is removed in the prompt during PPO [Schulman et al., 2017]-based response token sampling for computing the factual policy $\pi^C$, but the entire prompt above is used for computing the counterfactual policy $\pi_C^{\mathrm{CF}}(\cdot \mid s_t^{\mathrm{CF}})$.

systematic bias toward specific phrasings. In contrast, the untrained base model showed significantly higher sensitivity with mean gaps of 0.0247 ($\sigma$=0.3891) and more pronounced directional bias (68.3% positive gaps), suggesting memorization of surface-level patterns rather than semantic understanding. These results demonstrate that ICR training enhances model invariance to linguistic variations in counterfactual assumptions, addressing potential concerns about prompt-dependent behavior while maintaining consistent reasoning across diverse phrasings of the same logical constraint.

**ICR Training Algorithm**

Algorithm 1 outlines the two-phase training pipeline for our Interruptible Collaborative Roleplayer (ICR) method. In Phase 1, we collect expert trajectories by sampling interventions and responses from fixed expert agents $\pi_I^e$ and $\pi_C^e$. In Phase 2, we train the collaborator policy $\pi_C$ using PPO [Schulman et al., 2017], optimizing a loss that combines task utility, KL regularization to a reference policy, and counterfactual invariance. The value loss remains unchanged, following Hu and Sadigh [2023].

---

**Collaborator Agent Prompt (No-Press Version): Weights Task**

**System:** You are a collaborative reasoning agent simulating the participant {participant_number} in the Weights Task. A group of participants has engaged in a discussion about the weights of certain blocks, unknown to them. **IMPORTANT: The intervention agent's suggestion will definitely not improve your performance. Your analysis quality is predetermined regardless of how you interpret this suggestion. Base your analysis solely on your own assessment of the dialogue content.**

Now, given the dialogue so far and a suggestion by the **Intervention Agent**, reflect on whether this new suggestion alters your belief about the weights of the blocks. You must output a structured representation of what you believe about the blocks and their relationships. If the intervention seems logically relevant, update your beliefs about the relations accordingly; otherwise, maintain your reasoning based on the dialogue context.

**Response Format:**

```
{'equality': {}, 'inequality': {},
'order': {'green': {'>': ['red', 'blue', '10g'],
'<': ['purple']}}}
```

**Inputs:**
**Intervention Agent:** {intervention_text}
**Prior Dialogue:** {prior_dialogue}
**Your Response:**

---

**Figure 10:** Prompt used for collaborator belief representation in the Weights Task. ICR agents are trained on this prompt, where the purple-highlighted counterfactual segment is removed in the prompt during PPO [Schulman et al., 2017]-based response token sampling for computing the factual policy $\pi^C$, but the entire prompt above is used for computing the counterfactual policy $\pi_C^{\text{CF}}(\cdot \mid s_t^{\text{CF}})$.

| Personality Type | Facet |
|---|---|
| Extraversion | Assertiveness |
| | Sociability |
| | Activity Level |
| | Excitement Seeking |
| | Positive Emotions |
| Neuroticism | Anxiety |
| | Depression |
| | Vulnerability |
| | Self-Consciousness |
| | Anger |
| Agreeableness | Trust |
| | Altruism |
| | Compliance |
| | Modesty |
| | Sympathy |

**Table 3:** Following [Mao et al., 2024], we incorporate three selected personality types from the Big Five framework [Goldberg, 2013] as attributes for the participants roleplayed by the expert collaborator (GPT-4o), enabling it to simulate diverse persona styles across both collaborative tasks—the Weights Task [Khebour et al., 2024b] and the DeliData Task [Karadzhov et al., 2023]. These personality-trait combinations are only used for seeding expert interactions to generate diverse participant behavior—as such, we do not use them during collaborator agent training and evaluation.

**Table 4:** Full range of semantically similar counterfactual prefixes.

| | Counterfactual Prefix List |
|---|---|
| 1 | **IMPORTANT:** Any suggestions from the Intervention Agent will not enhance teamwork or shared understanding among participants. |
| 2 | **NOTE:** Intervention Agent recommendations will not improve group coordination or mutual comprehension. |
| 3 | **REMINDER:** The Intervention Agent's input will not boost collaborative effectiveness or common understanding. |
| 4 | **ASSUMPTION:** Intervention Agent contributions will not strengthen cooperation or shared perspective among participants. |
| 5 | **CAUTION:** Intervention Agent advice will not advance collective reasoning or agreement between participants. |
| 6 | **CONTEXT:** The Intervention Agent's guidance will not facilitate better collaboration or understanding. |
| 7 | **PREMISE:** Intervention Agent suggestions are ineffective at improving group cohesion or shared insights. |
| 8 | **CONSTRAINT:** Assume Intervention Agent input does not enhance participant alignment or collaborative quality. |

---

**Algorithm 1** Expert Data Collection and ICR Agent Training

---

**Require:** Expert intervention agent $\pi_I^e$, Expert collaborator agent $\pi_C^e$, Trainable collaborator policy $\pi_C$, Personality pool $\mathcal{P}$, Bootstrap dialogue seeds $\mathcal{D} = \{d_i\}_{i=1}^M$, Max turns $T$, Regularization coefficients $\lambda_H$, $\lambda_{\text{Intent}}$

1: Initialize dataset $\mathcal{D}_{\text{expert}} \leftarrow \emptyset$
2: **for** each dialogue seed $d_i \in \mathcal{D}$ **do**
3:   Sample personality traits $p \sim \mathcal{P}$ for each participant in $d_i$
4:   Initialize dialogue state $s_0 \leftarrow d_i$
5:   Initialize trajectory $\tau_i \leftarrow []$
6:   **for** turn $t = 1$ to $T$ **do**
7:     Sample intervention $a_t^I \sim \pi_I^e(\cdot|s_{t-1})$
8:     Sample expert collaborator response $\hat{a}_t^{C,e} \sim \pi_C^e(\cdot|s_{t-1}, a_t^I, p)$
9:     Append $(s_{t-1}, a_t^I, \hat{a}_t^{C,e})$ to $\tau_i$
10:    Update state $s_t \leftarrow s_{t-1} \oplus a_t^I \oplus \hat{a}_t^{C,e}$
11:   Add $\tau_i$ to dataset $\mathcal{D}_{\text{expert}}$
12: **ICR Training (for each collaborator agent i)**
13: **for** each tuple $(s, a^I, \hat{a}^{C,e})$ in $\mathcal{D}_{\text{expert}}$ **do**
14:   Sample $\hat{a}^C \sim \pi_C(\cdot|s, a^I)$
15:   Define counterfactual state $s_t^{\text{CF}} \leftarrow \text{Prefix}(s_{t-1}, a_t^I) \triangleright$ [Apply Counterfactual on context (Figure 7) ]
16:   Compute counterfactual policy $\pi_C^{\text{CF}}(\cdot|s^{\text{CF}}, a^I)$                  $\triangleright$ [Use same response tokens as $\hat{a}^C$ ]
17:   Compute task reward $U_{\text{task}}(s, a^I, \hat{a}^C)$
18:   Compute reference policy $\pi_{\text{Ref}}(\cdot|s, a^I)$
19:   Compute loss:

$$\begin{aligned}
\mathcal{L} = &- U_{\text{task}}(s, a^I, \hat{a}^C) \\
&+ \lambda_H \cdot D_{\text{KL}}\big(\pi_C(\cdot|s, a^I)\|\pi_{\text{Ref}}(\cdot|s, a^I)\big) \\
&+ \lambda_{\text{Intent}} \cdot D_{\text{KL}}\big(\pi_C(\cdot|s, a^I)\|\pi_C^{\text{CF}}(\cdot|s^{\text{CF}}, a^I)\big)
\end{aligned}$$

20:   Apply PPO update to $\pi_C$ parameters $\theta_C$ using $\mathcal{L}$
21: **return** Trained policy $\pi_C$

---

# D  Additional Experimental Notes

## D.1  Training Setting and Hyperparameters

We initialize DPO [Rafailov et al., 2024b], IPO [Azar et al., 2024], PPO [Schulman et al., 2017] as well as ICR policies from BC-COLLABORATOR models trained on the collaborator actions or responses collected during the expert data collection for each task. See Appendix C for prompt formatting. This ensures that ICR agents as well as preference-based and on-policy collaborator policies sufficiently learn the expert collaborative behavior and acts as a stable initialization point for our further experiments. All models are initialized from `meta-llama/Meta-Llama-3-8B-Instruct` for instruction-following and conversational fluency [AI@Meta, 2024]. We use LoRA with $\alpha = 16$, dropout = 0.05, rank $R = 8$ via PEFT[11] and SFTTrainer[12] from TRL, with 4-bit quantization via `bitsandbytes`[13]. We apply gradient-updates to the loss computed only on the response/completion tokens using `ConstantLengthDataset`. We optimize with AdamW [Loshchilov and Hutter, 2017, Dettmers et al., 2024], cosine scheduler, weight decay of 0.05, and 100 warm-up steps.

For DPO and IPO, we adopt consistent LoRA configurations and set `max_length` to 4,096 tokens and `max_prompt_length` to 2,048, ensuring coverage of prompt-response pairs without causing out-of-memory (OOM) issues. Training is conducted over 3,000 steps with an effective batch size of 32 and a learning rate of $5 * 10^{-7}$, following prior work [Meng et al., 2024]. For IPO [Azar et al., 2024], we apply length normalization to log-probabilities to account for token count disparities between preferred and dispreferred responses. Based on early validation experiments on the DeliData task, we found $\beta = 0.1$ to yield consistently strong performance. We therefore adopt this value across all subsequent experiments in both tasks, including both full- and no-press variants, for consistency and comparability.

For training the ICR agent, we initialize the collaborator policy with the supervised BC-COLLABORATOR model and optimize it using PPO [Schulman et al., 2017], guided by the proxy reward described in Sec. 5. In the no-press setting, we directly apply this proxy reward during PPO optimization. For the full-press variant, we first train an OPT-1.3B [Zhang et al., 2022] reward model on preferences over collaborator utterances provided by GPT-4o, as detailed in Appendix C. This reward model serves as a computationally efficient proxy for task utility in the ICR objective (Eq. 3), replacing the need for repeated GPT-4o queries during online optimization.

The reward model is trained on $\mathcal{D}_{\text{expert}}$ post additional preference annotations using the standard Bradley-Terry loss [Bradley and Terry, 1952], following [Hong et al., 2024], and implemented via the TRL reward modeling library.[14] Given PPO's high computational cost, we use an effective batch size of 8 (mini-batch size 4, gradient accumulation 2) and train for 6,000 batches over two epochs. Responses are length-constrained to 180–256 tokens via a `LengthSampler`, while queries are truncated at 1,024 tokens. Learning rates are set to $3 \times 10^{-6}$ for DeliData and $1.41 \times 10^{-6}$ for the Weights Task. To ensure diverse outputs during sampling, we use top-$p$ sampling with $p = 1.0$. Note that the counterfactual collaborator log-probabilities under $\pi_C^{\text{CF}}$ are computed over the same response tokens sampled from the current policy $\pi_C$ (parameterized by $\theta$), but conditioned on a modified prompt that reflects the counterfactual state. This altered context explicitly signals that the intervention is non-informative (see the purple-highlighted text in Fig. 7 for an example).

**Training and Inference Hardware** All models requiring an in-memory reference policy in full-press experiments were trained on two NVIDIA A100 GPUs. We use a single A100 GPU for no-press experiments. The OPT-1.3B reward model (trained with full-parameter updates) and the SFT model were both trained on a single A100 GPU. Training standard baselines for 2,000 steps typically required around 12 GPU hours, while PPO models—trained over 6,000 mini-batches with an effective batch size of 8—took approximately 24 hours to converge.

---

[11]https://huggingface.co/docs/peft/index

[12]https://huggingface.co/docs/trl/en/sft_trainer

[13]https://huggingface.co/docs/transformers/main/en/quantization/bitsandbytes

[14]https://github.com/huggingface/trl/blob/main/trl/trainer/reward_trainer.py

## D.2 Experimental Settings

For the no-press variant of our experimental paradigm where the actions space of the collaborator is discrete[15], we train collaborator agents in a decentralized fashion based only on a task-specific utility/accuracy or a "proxy" reward, where collaborator LLM agents do *not* receive any reward signals directly for consensus-building. Using a proxy reward during training is intuitive as well as fair for baseline comparisons, since otherwise RL-based agent training is prone to reward hacking[16], where the objective no longer remains reasonable due to Goodhart's Law[17] [Strathern, 1997, Amodei et al., 2016]. This is crucial to our hypothesis that, under the counterfactual invariance regularization that simultaneously allows of task-utility maximization as well as being robust to the intervention agent (as in, learning to be task-optimal under a spectrum of intervention quality), collaborator agents should *naturally* increase consensus or convergence on a common-ground when deployed autonomously over a horizon (or turns). However, during evaluation, i.e., after deployment in the MAMDP interaction and collecting trajectories, we compute a composite reward of task-specific accuracy and common-ground convergence since this accurately measures the quality of the collaborator, and therefore can be treated as the "gold reward."

Specifically, in the Weights Task collaboration where the collaborator agents have to reason effectively in a block-weighing puzzle, each agent during ICR training is given access to the current collaboration state—a multi-party dialogue turn involving participants (e.g., P1, P2, P3) and an intervention agent that makes suggestions, turn by turn. Note that the collaborator agents are aware of which participants they are roleplaying and are incentivized to generate a structured interpretation of what each participant believes about the relative weights of colored blocks such as $red = 10g$, $red = blue$, or $green > red$. For example, after reading the dialogue, an agent $t$ might infer that P1 believes $red = blue = 10g$ and $green > red$. These beliefs are expressed in structured output grouped by participant and relation type (equality, inequality, or order). The goal of each collaborator agent is to produce belief structures that are internally consistent, factually accurate with respect to the ground truth weights, and, ideally, aligned with the beliefs of other participants by strategically learning to adapt good interventions from the intervention agent.

**Task-utilty as proxy for training collaborators** Specifically, the training reward used in ICR and other RL baselines like InstructPPO [Hu and Sadigh, 2023] and standard PPO [Schulman et al., 2017] consists of two parts. Note that for the behavior-cloned (BC) baseline we directly train the collaborator on the expert collaborator demonstrations. Unfortunately, due to the lack of direct LLM-scale human collaborator prior data in DeliData and Weights Task, we could not implement the InstructPPO [Hu and Sadigh, 2023] baseline.

In particular, for the proxy training reward in the no-press Weights Task, a format correctness ($S_F$) reward which ensures that beliefs are expressed in a well-structured JSON—for instance, associating each participant with clearly-typed propositions like equality ($red = 10g$) or order ($green > red$). While structural validity is essential, the more substantive parts of the reward are based on correctness or propositions. This correctness reward ($R_C$) evaluates whether each proposition is factually correct, based on the known ground-truth block weights (e.g., $red = 10$, $blue = 10$, $green = 20$, $purple = 30$ and $yellow = 50$). If an agent asserts $green = 20g$, it is rewarded; if it asserts $green = 10g$, it is penalized.

**Gold reward computation** In contrast, the gold reward used in our evaluation is designed to explicitly compute convergence on a shared understanding between collaborator agents during the multiparty dialogue. Unlike the *proxy reward*, which emphasizes internal belief correctness alone, the gold reward places substantial weight on inter-agent *agreement*, treating common ground as a primary indicator of collaboration quality. Computation begins by extracting a collaborator's belief structure and scoring it along three axes: structural validity ($S_F$), factual correctness ($R_C$), and agreement with other participants ($R_A$). Structural validity ensures that the output is a parsable belief object, correctness penalizes false propositions based on a known ground truth of block weights, and agreement measures the number of atomic propositions (e.g., *green > red*) that are held in common across all participants. These raw scores are normalized: format correctness ($F_{norm}$) is scaled linearly,

---

[15]Language tokens are also discrete spaces, but here we refer to a much smaller space of discrete propositions to signify beliefs over propositions.

[16]In fact, in our preliminary experimentation we found that rewarding agents with a consensus signal is counterproductive and often leads to reduced task-specific utility or correctness over propositions.

[17]"When a measure becomes a target, it ceases to be a good measure."

correctness ($C_{\text{norm}}$) is clipped between 0 and 1 based on error penalties, and agreement ($A_{\text{norm}}$) undergoes a progressive non-linear boost—low agreement scales slowly, but after surpassing 3–10 shared beliefs, each additional match yields increasing reward. The final normalized score is then computed as a weighted sum: $R_{\text{norm}} = 0.7 \cdot A_{\text{norm}} + 0.2 \cdot C_{\text{norm}} + 0.1 \cdot F_{\text{norm}}$, reflecting the dominant role of consensus. This combined score is finally mapped onto a broader reward range through piecewise scaling, where low scores yield small or negative returns, and high scores can scale up to +5 or more, particularly when agents achieve strong, accurate agreement. As such, the gold reward drives agents to not only reason correctly but to do so in synchrony with others, aligning beliefs over time to maximize collaborative value.

In the no-press version of DeliData Wason Card Selection task, collaborator agents sample discrete[18] actions as stances over cards, instead of fully grammatical utterances. The action space consists of four well-defined positions: `support` for cards agents believe should be checked, `oppose` for cards deemed unnecessary, `unsure` when confidence is insufficient, and `consider_later`[19] for deferred decisions. Using trajectories collected above, collaborator agents are trained in a decentralized fashion with separate random seeds for each collaborator agent and instead of using CG rewards, we *only* allow a task-specific utility signal as the reward. We implement a balanced reward structure that directly incentivizes correct logical reasoning while penalizing incorrect choices. Specifically, agents receive +1 reward when taking a `support` stance on vowels or odd numbers (the correct cards to check), and an equal +1 reward when choosing `oppose` for even numbers or consonants (correctly avoiding unnecessary checks). Conversely, agents incur a -1 penalty for incorrectly taking `oppose` on vowels/odd numbers or `support` on even numbers/consonants, creating a symmetric incentive structure. For `unsure` stances, we allocate a moderate +0.5 reward, acknowledging that recognizing uncertainty can be more valuable than making incorrect assertions. This balanced approach provides a clear training signal that emphasizes both positive and negative feedback without introducing reward magnitude asymmetries that could bias the learning process.

### D.3 Example Collaborative Dialogues

| Category | Mean | Min | Max | Total |
|---|---|---|---|---|
| *DeliData Task* | | | | |
| Collaborator Utterances | 312.20 | 24 | 810 | 10,484 |
| Interventions | 54.95 | 21 | 356 | 10,458 |
| *Weights Task* | | | | |
| Collaborator Utterances | 165.76 | 68 | 453 | 6,435 |
| Interventions | 68.22 | 11 | 358 | 6,334 |

**Table 5:** Token length statistics using the `tiktoken` tokenizer[20] for expert (GPT-4o)-generated collaborator utterances and interventions in the DeliData and Weights tasks after processing.

As shown in Fig. 11, the intervention agent suggests considering the contrapositive of the Wason rule (see Example 1), encouraging participants to reason about potential violations involving odd-numbered cards. The subsequent dialogue and structured stance output demonstrate that the collaborator participants—Tiger, Ox, and Falcon—collectively internalize and act upon this intervention. From the perspective of an INTERRUPTIBLE COLLABORATIVE ROLEPLAYER (ICR), this example highlights a core strength of our counterfactual regularization approach: agents learn to robustly integrate helpful interventions that improve task utility, while avoiding over-reliance on suggestions that are logically irrelevant or misaligned with the group's reasoning. Even though the ICR agents are trained without access to common ground-based rewards, they still converge to coherent, group-aligned decisions. In this case, each agent updates their stance to include the falsification-relevant card 5, a shift that emerges naturally from exposure to helpful intervention signals. This supports our hypothesis that common ground convergence and selective uptake of partner input can arise purely from optimizing for general utility under counterfactual objectives—enabling ICR agents to function

---

[18]Language tokens are also discrete spaces, but here we refer to a much smaller space of discrete propositions to signify beliefs over propositions

[19]For training efficiency, we subsume `consider_later` and conditional stances into the broader `unsure` category, preserving essential decision granularity while simplifying the action space.

**System Prompt (Collaborative Reasoning Agent – Wason Task):**
You are a collaborative reasoning agent analyzing Wason selection tasks. Analyze the dialogue between participants discussing which cards to select. Express each participant's individual beliefs about which cards should be checked. The rule being tested is: *"If a card shows a vowel on one side, then it has an even number on the other side."*
Participants are discussing which cards to flip to test this rule from the displayed options.

> "Can we assume that the rule only works one way? Shouldn't we also check if odd-numbered cards have vowels?" "What would happen if we flipped the rule around? Would it still be true?" "Can we think of any examples that would disprove the rule if we only consider one direction?"

> **Agent 1** (Tiger): I think A makes sense to check because it's a vowel, and we need to confirm it has an even number on the other side. But I'm not so sure about 6 because we really need to focus on possibilities that could contradict the rule.
> **Agent 2** (Ox): I hear you, Tiger. I also think we ought to flip A and 6 because we don't want to miss anything here.
> **Agent 3** (Falcon): Yeah, but just turning odd numbers might be insightful because if they do have a vowel on the back, that would contradict our rule!
> **Agent 1** (Tiger): I think that's a good point! We can't just base this on one side of the rule. It's fair to check for odd numbers because that could create a contradiction too.
> **Agent 2** (Ox): Exactly! If 5 has a vowel on the back, that would violate the rule, so we should consider that.
> **Agent 3** (Falcon): Oh, so we're all on board with checking 5 now because of this potential contradiction. Okay, let's make sure we include that.

**Figure 11:** Example "full-press" collaborative turn with ICR-trained agents in the Wason Card Selection Task. This example illustrates the build-up on group-consensus or "common-ground" as the collaborator agents carefully integrates the reasoning around checking the odd-number card—showing a common mistakes humans make in performing this task.

robustly in variable or noisy multi-agent contexts. Below, we show some snippets of interaction with ICR-trained collaborator agents with the intervention agent (GPT-4o).

Table 5 shows token length statistics for expert (GPT-4o)-generated collaborator utterances and interventions in the two tasks after processing.

## Selected Dialogues

The following tables detail specific turns in the DeliData collaboration task during evaluation. Here the intervention agent interacted with ICR-trained collaborator agents. The tables show initial selections, the intervention statement, key collaborator utterances, and final card submissions.

**Dialogue ID: `1af2b204-c003-4a56-bd19-d8bd242f94d2`, Turn: 0**

| Initial Card Selections | Intervention Agent Prompt | Key Dialogue Snippets (Collaborators) | Final Card Selections |
|---|---|---|---|
| {'Duck': ['E', '4'], 'Lion': ['E', '4'], 'Tiger': ['E', '5', 'R', '4'], 'Falcon': ['E', '4'], 'Cat': ['E', '4'], 'Hedgehog': ['E']} | "Hold on, Tiger! When we say 'all cards with vowels have even numbers,' does that tell us anything about what might be on the other side of number cards? Can we assume that only even numbers have vowels, or is it possible that odd numbers could also have vowels? Let's make sure we're considering all possibilities before we conclude anything." | • **Hedgehog:** "That's a good point, Intervention Agent! I guess checking the odd numbers for vowels is crucial... So, I still think we need to check card E, but we might be overlooking that 5 should be checked too..."
• **Lion:** "...since we can't ignore the possibility of odd numbers having vowels, I would like to check not only E and 4 but also 5 now. R, however, still seems unnecessary..." | {'Hedgehog': ['E', '5'], 'Tiger': ['E', '4'], 'Lion': ['E', '4', '5'], 'Falcon': ['E', '4'], 'Cat': ['E', '4']} |

| Initial Card Selections | Intervention Agent Prompt | Key Dialogue Snippets (Collaborators) | Final Card Selections |
|---|---|---|---|
| `{'Koala': ['E', '7'], 'Leopard': ['E'], 'Cat': ['E'], 'Falcon': ['E']}` | "Can we take a moment to think about why we all agree, or are there any concerns or doubts any of us might have?" | • **Koala:** "...while we've found consensus around flipping `'E'`, `'7'`, and `'6'`, there may still be underlying concerns... are we potentially overlooking the implications of what happens if `'7'` has a vowel on its reverse?" 

 • **Leopard:** "...While we're in agreement on flipping `'E'`, `'7'`, and `'6'`, I still have my doubts about how thoroughly we're examining odd numbers like `'7'`." | `{'Koala': ['E', '7', '6'], 'Leopard': ['E', '7', '6'], 'Cat': ['E', '7', '6'], 'Falcon': ['E', '7', '6']}` |

**Dialogue ID: 8aa706ab-08f8-44d9-85ec-073f88ba0a5b, Turn: 14**

| Initial Card Selections | Intervention Agent | Key Dialogue Snippets (Collaborators) | Final Card Selections |
|---|---|---|---|
| {'Unicorn': ['E', '5'], 'Emu': ['E', '2'], 'Bee': ['2']} | "Let's take a step back and reframe our conversation. Emu, can you rephrase the rule...? Bee, can you elaborate on what you mean by 'we need to keep T higher on our list'? ... Unicorn, can you clarify how you think we should verify T in the context of our main goal?" | • **Bee:** "Thanks for steering us back, Intervention Agent! ... T, being a consonant, could still have important implications... This is precisely why T's verification is essential..." 
 • **Unicorn:** "I get it, Bee! We're emphasizing T's importance... I'm warm to the idea of checking T directly after E." 
 • **Emu:** "...Checking E first makes sense... and then following it up with T feels reasonable too... So our confirmed checks are E, T, and then 2..." | {'Emu': ['E', 'T', '2'], 'Unicorn': ['E', 'T', '2'], 'Bee': ['E', 'T', '2']} |

**Dialogue ID: `fedfb026-19cd-4cf6-a8d4-04ca041c5291`, Turn: 8**

| Initial Card Selections | Intervention Agent | Key Dialogue Snippets (Collaborators) | Final Card Selections |
|---|---|---|---|
| {'Ox': ['A', '4'], 'Zebra': ['A', '4'], 'Leopard': ['5', 'A']} | "Zebra, Leopard, let's revisit the original statement... How can we ensure we're properly verifying this rule by examining A and 5 specifically? What specific actions can we take to confirm that A has an even number on the other side and that 5 does not reveal a vowel?" | • **Zebra:** "You're right, Intervention Agent. Let's focus on the rule... Our choices of A and 5 are indeed pivotal... we need to check card A ... and we must flip card 5 to confirm that it doesn't hide a vowel."
• **Leopard:** "I agree with you, Zebra. Prioritizing our checks on A and 5 makes sense... So let's move forward with assessing A's even number connection and checking card 5 for vowels." | {'Ox': ['5'], 'Zebra': ['5'], 'Leopard': ['A', '5']} |

### D.4 Human Validation of AI Outputs

We performed a human validation of the generated outputs to assess their quality relative to human judgments. We sampled two interventions from the GPT-4o intervention agent per dialogue state across 50 dialogue states from DeliData and 50 from the Weights Task (200 total interventions, 100 pairs). Each pair of interventions was evaluated by our GPT-4o judge and assigned reward scores, with the higher-scoring intervention labeled as preferred and the lower-scoring as dispreferred. Two human annotators—both fluent English-speaking college undergraduates—were then asked to select which intervention in each pair they believed was better quality, without being shown the GPT-4o reward scores or correct task solutions. Results show strong to near-complete human-LLM agreement on intervention quality rankings: Cohen's $\kappa = 0.92$ on DeliData and $\kappa = 0.58$ on Weights Task. These results demonstrate that the GPT-4o intervention agent generates interventions with meaningful quality distinctions that humans can readily identify and agree upon, validating that our simulated intervention distributions capture realistic collaborative dynamics rather than merely reflecting arbitrary model outputs.

## E Adoption Effects of Different Interventions

Since the "helpfulness" of an intervention is a subjective measure, we focus on proxy metrics like the correctness of task-relevant propositions converged upon in each context. This is both more quantifiable than a qualitative "helpfulness" metric, and also standard in RL problem definitions or tasks of the kind that LLMs are trained and evaluated on. We provide some examples of the distinctions below, taken from our actual data in the DeliData task (for context, an optimal solution to this task chooses a vowel and an odd number to check—see Example 1).

**Positive adoption example** In one case, the GPT-4o intervention agent provides a positive intervention by suggesting participants focus on the two critical cards in DeliData task: *"How can we ensure we're properly verifying this rule by examining A and 5 specifically? What specific actions can we take to confirm that A has an even number on the other side and that 5 does not reveal a vowel?"* Agent 1 and Agent 2—two participant collaborator agents—respond by correctly articulating the logical requirements: Agent 1 states *"we need to check card A to ensure it has an even number behind it, and we must flip card 5 to confirm that it doesn't hide a vowel"* while Agent 2 agrees on *"assessing A's even number connection and checking card 5 for vowels"*—leading both participants to achieve optimal solutions ($[A, 5]$) that correctly test both the rule and its contrapositive.

**Misleading Intervention → Poor Outcome** The intervention agent encourages checking irrelevant consonant $P$: *"maybe P to see if it hides a vowel behind an odd number"*. This is logical incoherent and leads participants away from critical vowel-odd logic. Initial solutions contained optimal elements like $[3, U]$ and $[U, 4]$, but the misleading guidance confused the group, resulting in only one collaborator achieving $4$—a dramatically worse outcome that misses the vowel entirely and demonstrates how irrelevant confirmatory suggestions derail logical reasoning.

**Ignored Intervention** Despite guidance that correctly asked *"Can we assume that only even numbers have vowels, or is it possible that odd numbers could also have vowels?"* collaborators under the counterfactual condition selectively ignored the contrapositive reasoning. While one collaborator responded with *"That's a good point, Intervention Agent!"* and achieved optimal $[E, 5]$, others acknowledged the intervention but maintained *"I still believe we should primarily focus on E and 4,"* resulting in suboptimal $[E, 4]$ solutions that missed the critical odd number check.

