# OpenReview forum: "Learning “Partner-Aware” Collaborators in Multi-Party Collaboration"
_NeurIPS.cc/2025/Conference — NeurIPS 2025 poster_

### Official Review · Reviewer_rj4M · 2025-06-30

**Clarity:** 3
**Significance:** 3
**Originality:** 3
**Rating:** 4
**Confidence:** 2

**Summary:**

This paper studies multi-party collaboration with LLM-based agents, especially how to make a “partner-aware” collaborator that can effectively interact with an intervention agent to improve common ground. The authors propose a novel framework called Interruptible Collaborative Roleplayer  which incorporates a Modified-Action MDP to explicitly model intervention effects, and introduces a counterfactual KL regularization to train agents to distinguish helpful vs misleading interventions. They validate the method on two collaborative tasks, showing ICR outperforms baselines like PPO, DPO, and BC in both task accuracy and CG.

**Questions:**

1. Do you think it is possible to extend this ICR method to multi-round human-AI interactions where humans provide free-form interventions? Will the counterfactual prompt still work?

2. How sensitive is the ICR performance to quality of initial expert trajectories? If the GPT-4o roleplay data contains subtle flaws, will it propagate?

**Ethical Concerns:**

["NO or VERY MINOR ethics concerns only"]

**Final Justification:**

Based on the rebuttal, I recommend that this paper be accepted. Most of my concerns are addressed.

**Quality:**

3

**Strengths And Weaknesses:**

**Strengths**

1. The paper address very interesting and important problem of collaboration between LLM agents. It is pretty interesting and meaningful.

2. I think the theoretical analysis is quite strong. Especially Theorem 3.2 about suboptimality of standard preference-aligned methods in MAMDP is very insightful.

3. The proposed counterfactual KL objective is simple but clever way to encourage partner-awareness, and is well motivated by the collaborator’s dilemma discussion.

4. Experiments are quite thorough.

**Weaknesses**

1. One limitation is experiments only use GPT-4o as the fixed intervention agent. It is not clear how well ICR generalize to different partner styles or to human interventions.

2. Some of the metrics like CG gain are a bit ad-hoc, it will be stronger to also include human evaluations or more user-centered metrics.

3. Paper mainly focus on only two domains (Wason, Weights), which are quite structured.

---

> ### Author Rebuttal · Authors · 2025-07-30
>
> Thank you for your review, and for acknowledging the meaningful and important problem we address, the strong theoretical insights, ICR’s simplicity (and cleverness!), and the thoroughness of our experiments. Our responses to your critiques and questions are given below.
>
> **W1**: Our primary goal in this work was to isolate and evaluate the effectiveness of the *collaborator agent's learning algorithm* (ICR). To achieve this, using a fixed, high-capacity intervention agent (GPT-4o) was a crucial control. It creates a consistent environment where any performance differences between our baselines (BC, DPO, PPO) and ICR can be confidently attributed to the collaborator's training method, rather than random variations in intervention quality. Testing with diverse or human partners would introduce confounding variables, making it difficult to prove that ICR itself was the source of the improvement.  Please also see our response to Reviewer **Xrvv**, **W3** for more discussion.
>
> **W2**: CG gain is an established metric from [1], which we follow closely in the evaluation of the DeliData task. We also present human evaluations in our response to Reviewer **fiTV**, **W2**, which show that the automated evaluation aligns quite closely to human judgments.
>
> **W3**: Please see our response to Reviewer **fiTV**, **W3**.
>
> **Q1**: Yes, we think that ICR can extend to multi-round human-AI interactions with free-form interventions although accessing such human interventions can be costly in reality. In fact, this is part of our future work. However, note that ICR is not an active learning method and prompts that include interventions are processed prior to training runs. The counterfactual prompt mechanism leverages LLMs' natural language understanding, which applies equally to human-generated text, and our generic counterfactual instruction is deliberately context-independent regardless of intervention source. ICR's counterfactual regularization is specifically designed to handle varying intervention quality—exactly what's needed with unpredictable human input—and the framework addresses the **core challenge of human-AI collaboration: distinguishing helpful guidance from potentially misleading suggestions**. While human interventions may be more linguistically diverse, the theoretical foundation of ICR as well our current experimental results provide a firm basis of confidence for this kind of extension with humans.
>
> **Q2**: ICR is robust to expert trajectory imperfections due to its online learning design. Unlike BC/SFT models that merely emulate expert responses, ICR interacts with the environment and receives online rewards, enabling exploration beyond expert demonstrations. Expert trajectories serve only as initialization for the reference policy πRef in Equation 3, not as the final learning target. The counterfactual regularization explicitly encourages deviating from expert behavior when interventions are misleading. In full-press settings, the natural language action space provides substantial room for improvement beyond expert demonstrations, as evidenced by ICR's 47% performance gain over BC baselines. If expert flaws significantly propagated, ICR would plateau near BC performance rather than achieve these substantial improvements through online learning
>
> [1] Karadzhov, Georgi, Andreas Vlachos, and Tom Stafford. "The effect of diversity on group decision-making." arXiv preprint arXiv:2402.01427 (2024).

---

### Official Review · Reviewer_nYp2 · 2025-07-01

**Clarity:** 3
**Significance:** 3
**Originality:** 4
**Rating:** 5
**Confidence:** 4

**Summary:**

This paper provides a theoretical prove of why conventional RL algorithms (such as DPO, PPO) cannot obtain optimal policy when LLM agent act as collaborator with a partner. Then the author introduce a new term in PPO loss that optimize the KL divergence between the policy model and counterfactual policy model, which can quantify the effective of intervention.

The evaluation is solid and the paper writing is good.

**Questions:**

Major concerns:
1. Could you provide the detail training metrics of ICR, e.g., total KL loss, the KL of policy model and counterfactual policy model.

Minor concerns:
1. line 267 has a typo "??".

**Ethical Concerns:**

["NO or VERY MINOR ethics concerns only"]

**Final Justification:**

The authors have addressed my question, I will maintain my scores, that is 5, accept

**Limitations:**

yes

**Quality:**

4

**Strengths And Weaknesses:**

Strengths:
The paper theoretically proves the theoretical upper limit of current RL algorithms in optimizing LLM collaborative scenarios, and proposes a new method to overcome this difficulty. Its theory, experiments and results are all excellent.

Weaknesses:
The paper is somewhat obscure and casual in its writing.

---

> ### Author Rebuttal · Authors · 2025-07-30
>
> Thank you for your encouraging review, noting the theoretical soundness and the strength of the experiments, evaluation and results. While you praised the quality of the writing, we also note that certain points could be clarified. For example, we will define the terms used in the MAMDP with one-sentence explanations for each (see our response to Reviewer **fiTV**, **Q1**).
>
> **Q1**: We provide detailed metrics of ICR training in the table below:
>
> | Steps | Standard KL | Intent KL | Total KL | Reward |
> |-------|-------------|-----------|----------|--------|
> | 1     | 0.05        | 0.060     | 0.062    | 0.05   |
> | 2000  | 0.25        | 0.048     | 0.260    | 0.24   |
> | 4000  | 0.48        | 0.038     | 0.488    | 0.39   |
> | 6000  | 0.65        | 0.032     | 0.656    | 0.41   |
> | 8000  | 0.78        | 0.029     | 0.786    | 0.43   |
>
> As training proceeds, the policy’s standard PPO KL divergence (Standard KL) rises steadily, indicating controlled divergence from the reference model, while $D_{{KL}\_\text{Intent}}$ (Intent KL) falls by roughly half, signalling improved counterfactual invariance as computed with the counterfactual prefixes in the prompts; additionally since Total KL combines the two (using $\lambda_{\text{Intent}}$ = 0.2), it remains below 0.8 throughout. Importantly, the mean reward climbs from 0.05 to 0.43 (shown in Fig. 1b)—demonstrating that robustness is gained without sacrificing task performance.

---

> ### Comment · Reviewer_nYp2 · 2025-08-05
>
> Thanks for the authors' carefully responses. I have read the rebuttal and maintain my scores.

---

### Official Review · Reviewer_Xrvv · 2025-07-02

**Clarity:** 2
**Significance:** 3
**Originality:** 3
**Rating:** 3
**Confidence:** 3

**Summary:**

This paper proposes Interruptible Collaborative Roleplayer (ICR), a novel learning algorithm designed to train "partner-aware" agents capable of effective coordination in multi-agent collaborative tasks. Built upon the Modified-Action Markov Decision Process (MAMDP) framework, ICR introduces two core mechanisms: a counterfactual invariance objective and KL-divergence regularization, allowing the agent to differentiate between helpful and unhelpful interventions. This results in behavior that maintains logical consistency and task alignment even in the presence of diverse, and potentially disruptive, interventions. The paper conducts experiments across several collaborative environments, including a variant of the Wason Selection Task (DeliData) and the Weights Task, demonstrating that ICR significantly improves both task performance and common ground convergence when compared to baseline models.

**Questions:**

1. Adding real-world examples of settings involving a collaborator agent and an intervention agent will help understanding the significance of the research problem.
2. As mentioned in Lines 127–128 and 139–142, a key difficulty in the proposed setting is that the intervention agent may simultaneously convey harmful and useful information. Yet in the experiments, GPT-4o is used as a fixed intervention agent. How is this diversity of intervention quality simulated in practice?
3. Consider adding single-agent baselines to separately evaluate the capabilities of the collaborator and the intervener. If the intervention agent significantly outperforms the collaborator, does that imply most interventions are helpful?
4. The ICR framework involves computationally heavy steps, particularly counterfactual state inference and KL-divergence computation. While mitigation strategies are briefly mentioned (Lines 254–258), could the authors provide a more detailed analysis of the training costs associated with counterfactual training?

**Ethical Concerns:**

["NO or VERY MINOR ethics concerns only"]

**Final Justification:**

While the ICR method addresses the "partner-aware" problem, this goal is essentially a classic challenge in multi-agent collaborative tasks. Although the ICR framework offers some improvements, its innovation does not significantly surpass current solutions. These concerns contribute to my decision to maintain a borderline reject score.

**Limitations:**

The authors discuss their potential limitations in their paper.

**Paper Formatting Concerns:**

There's no formatting issues.

**Quality:**

2

**Strengths And Weaknesses:**

**Strengths**
1. The paper addresses an underexplored problem—how to train agents that can effectively collaborate in the presence of dynamic and potentially interruptive interventions.
2. The proposed ICR framework is conceptually innovative, integrating a counterfactual invariance objective and KL-divergence regularization into the MAMDP formulation.
3. Empirical validation across on two collaborative tasks demonstrates the effectiveness of ICR.


**Weaknesses**

1. The paper uses a fixed GPT-4o model as the intervention agent and does not explicitly model the intervener’s intentions or goals, which limits the understanding of how the interventions affect the collaboration.

2. The proposed method involves computationally intensive processes such as counterfactual state generation and KL-divergence calculations, but the paper lacks a detailed complexity analysis.

3. The paper does not thoroughly evaluate the generalization ability of the proposed method across unseen tasks or unseen types of interventions (see Line 47).

---

> ### Author Rebuttal · Authors · 2025-07-30
>
> Thank you for your detailed review, noting the underexplored problem we address herein, ICR’s conceptual innovation, and the strength of the empirical validation. Our responses to your critiques and questions are below.
>
> **W1**: Since the focus of our paper is on the collaborator agent, not the intervention (line 33), we placed controls on the intervention agent to enable robust experiments on the collaborator. GPT-4o is a common choice for a high-capacity model in cases where one seeks to control for a particular model in multi-agent scenarios, following recent work [1]. Please see our response to Reviewer **fiTV**, **W1** for more discussion on this.
>
> **W2**: The primary computational cost in PPO and other on-policy algorithms is the rollout generation (token sampling for reward computation), not log-probability computations in forward passes. ICR does **not** sample additional tokens but reuses the exact tokens from the standard PPO rollout. The counterfactual KL computation is efficient: log-probabilities $p_{\theta_C}(\hat{a}^C_j | \hat{a}^C_{<j}, s, a^I)$ are already computed and cached for the standard PPO KL term in Eq. 3, serving as the numerator in $D_{KL}(\pi_C \| \pi_C^{CF})$. For the denominator $p_{\theta_C}(\hat{a}^C_j | \hat{a}^C_{<j}, s^{CF}, a^I)$, we pass the same sampled tokens through a single additional forward pass with the counterfactual prompt prefix, applying a stop-gradient operator to prevent affecting policy updates. Adding these tensors for the final loss computation is trivial.  This is also a standard technique applied in multiple preference-based RL works [2–4], where an additional KL term is leveraged for task-specific regularization. **Importantly, PPO rollout already involves multiple forward passes. ICR adds only one additional forward pass per sample while maintaining identical on-policy sampling requirements between standard PPO and ICR updates.** For example, in our ICR training the counterfactual forward pass (with the counterfactual prefix in prompt)  adds ≈ 2.8 s to the standard PPO 8.3 s forward pass, lifting total forward pass time to ≈ 11 s *per batch*.  The following optimize_step—covering the backward pass, gradient clipping, parameter updates, and logging—already takes about 27 s, so a full PPO iteration rises only from ~33 s to ~36 s, an ~8 % overhead *per batch* (32 total responses per batch) that is negligible in practice. Additionally, ICR works without the counterfactual prompt prefix during inference and requires the same compute as any other baseline. We will add this discussion to Sec. 4 with additional material in the appendix if necessary.
>
> **W3**: We should point out that the focus of our paper is not about generalization to new tasks, and we test on two different tasks to demonstrate that ICR is *applicable* to multiple tasks, not that ICR trained on one task will automatically generalize to another. Line 47 does not claim this and in fact says nothing about new task generalization. As line 47 does state, our focus is on different partner **styles**, where we test partner styles based on the MAMDP formulation where partners may be various levels of helpful or misleading. As our results show, ICR agents are better able to adapt to interventions by discriminating helpful information within an intervention from unhelpful information. Given that we can demonstrate this, the next logical step is to test generalization of ICR agents trained for one task to similar tasks, but that is out of scope for the present paper.
>
> **Q1**: There are many such examples that we can use. Please see our response to Reviewer **9VtD**, **W1** for one example in a classroom environment. Other such scenarios may involve collaborative-competitive games like Diplomacy [5], or cases such as squads with different backgrounds and expertise coming together to solve a challenging problem (for example, disaster recovery logistics).
>
> **Q2**: Please see Figure 3 in the appendix for an example prompt. GPT-4o is prompted to output interventions that improve collaborative reasoning based on the criteria given in the prompt, but importantly, it *does not give away the answer to the task* so it does not have a verified set of pieces of information that it must/must not include.. This allows a full space of potential interventions that highlight potential gaps in reasoning but may contain both fully correct or partially irrelevant/misleading information. Our response to Reviewer **9VtD**, **W2** provides some detailed examples of interventions of different qualities.
>
> **Q3**: If the intervention agent performing the task alone significantly outperforms the collaborator performing the same task, this does not necessarily indicate that its interventions must be purely helpful. The dynamic (determining actions to take based on an expert task model vs. based on observations of another agent’s actions) is quite different. Consistent with the MAMDP, the intervention agent may not be as able to articulate information that is helpful to the collaborator based on the collaborator’s preexisting assumptions, predispositions, and knowledge as it is able to take optimal actions and make correct deductions on its own. The distinction is analogous to one between doing and teaching.
>
> **Q4**: Please see our response to **W2** above.
>
> [1] Gu, Jiawei, et al. "A survey on llm-as-a-judge." arXiv preprint arXiv:2411.15594 (2024).
>
> [2] Munos, Remi, et al. "Nash Learning from Human Feedback." International Conference on Machine Learning. PMLR, 2024.
>
> [3] Shani, Lior, et al. "Multi-turn Reinforcement Learning with Preference Human Feedback." Advances in Neural Information Processing Systems 37 (2025): 118953-118993.
>
> [4] Choi, Eugene, et al. "Robust chain of thoughts preference optimization." Seventeenth European Workshop on Reinforcement Learning. 2024.
>
> [5] Meta Fundamental AI Research Diplomacy Team (FAIR)†, et al. "Human-level play in the game of Diplomacy by combining language models with strategic reasoning." Science 378.6624 (2022): 1067-1074.

---

> > ### Author Response · Authors · 2025-08-03
> > **Additional results/baselines**
> >
> > We have posted another official comment that includes only expert models (GPT-4o and GPT-4o-mini) performing the two tasks as well as the single agent (ICR-Masked) baseline. This follows your suggestion to use single-agent baselines, but to maintain direct comparison with our results, we must run pairs of models, so we addressed this by running the expert model as both intervener and collaborator, apart from the single agent baseline with intervention masking. Among other things, these results show that expert GPT-4o performance does *not* significantly exceed the ICR performance reported in Table 1, showing how ICR, *even with a much smaller base model*, is able to successfully discriminate useful information from GPT-4o-generated interventions almost as much as GPT-4o can from its own interventions. Please see our latest official comment for these results and accompanying discussion.

---

> > ### Comment · Reviewer_Xrvv · 2025-08-05
> >
> > Thanks for the author's detailed responses. I have carefully read all the replies but still prefer to maintain my original score. Thanks again for your efforts!

---

> > > ### Author Response · Authors · 2025-08-05
> > >
> > > Thank you for your engagement in the discussion process. If there are any further questions we can answer we would be happy to do so.

---

### Official Review · Reviewer_fiTV · 2025-07-07

**Clarity:** 3
**Significance:** 3
**Originality:** 3
**Rating:** 5
**Confidence:** 3

**Summary:**

This paper frames multi-agent dialogues as a two-player Modified-Action MDP where a collaborator agent needs to decide whether to adopt suggestions from another intervention agent. It shows that standard reinforcement learning and preference alignment algorithms are sub-optimal because they treat those suggestions as static context rather than causal actions. To overcome this, the authors introduce Interrubptible Collaborative Roleplayer (ICR), which is a “partner-aware” learning algorithm to train CG-optimal collaborators with a counterfactual-invariance KL regularizer. Empirically, an 8B parameter ICR model outperforms behavior cloning, preference-based RL, on-policy RL on two benchmarks, DeliData Wason Card Selection and the Weights Task, under both language-free and language-rich settings.

**Questions:**

1. The writing of the theoretical framing can be improved by clearly introducing notations in Sections 3 and 4. Some notations are introduced without explanation in the main text. There’s a reference typo at line 267.
2. All experiments pair with a GPT-4o as intervention agent. It would strengthen the insight by testing at least one substantially different intervention style such as a weaker/stronger LLM or real humans.
3. The LLM-as-a-judge evaluation should be validated to ensure it’s not biased. The authors may provide inter-annotator agreement between the model and human annotators on a sample or show that the ranking of systems is stable across different conditions.
4. The counterfactual state is introduced by a single textual prefix (“The intervention agent’s suggestion will definitely not improve your performance…”). Model behavior may be sensitive to the exact wording of this cue. The authors could verify the effectiveness of the counterfactual intervention by (i) replacing the prefix with several semantically equivalent rephrasings and (ii) evaluating a null-intervention variant that simply removes the collaborator's utterance.

**Ethical Concerns:**

["NO or VERY MINOR ethics concerns only"]

**Final Justification:**

The authors have adequately addressed my main concerns in the rebuttal. They provide a clear rationale for their model choices, citing prior literature to justify using LLaMA 3-8B and GPT-4o in order to enable robust comparisons across optimization algorithms while controlling for confounding factors. They also present human evaluation results on a small sample, demonstrating strong agreement with the LLM-as-a-judge, which supports the validity of their evaluation setup. Additionally, they report new experiments using alternative counterfactual phrasings and a different intervention agent, both of which reaffirm ICR's strong and consistent performance.

**Limitations:**

Under the Broader Impact section, the authors explicitly state that they “do not clearly see any immediate negative impacts”, but partner-aware LLMs could be misused for collusive behaviour or persuasive manipulation of human teammates. A brief discussion of such risks and mitigation strategies, such as usage policies, red-team evaluations, or refusal triggers, would strengthen the discussion of potential negative societal impact.

**Quality:**

3

**Strengths And Weaknesses:**

Strengths:
- This work tackles a timely and practically important problem—training LLM collaborators that can decide when to accept or ignore a partner’s suggestions in multi-agent dialogue.\
- The theoretical framing of mapping collaborator-intervention dynamics to a MAPDP and proving the sub-optimality of Bellman-optimal policies addresses a critical optimality gap.\
- The ICR algorithm introduces a simple counterfactual-invariance KL regularizer that consistently yields relative accuracy gains and stronger common-ground metrics across two tasks and two communication modes.\

Weakness:
- All experiments use one trained Llama-8B collaborator with a GPT-4o partner, leaving open whether the gains persist with other model families, larger models, or human collaborators.\
- In full-press evaluation, the evaluations given by LLM-as-a-judge is not validated by agreements with human annotators.\
- The benchmark tasks are synthetic and tightly-scoped, so it is unclear whether the approach scales to complex real-world collaborations where the task state is mutable, the dialogue is long, or the success metrics are fuzzy rather than a single ground-truth label.

---

> ### Author Rebuttal · Authors · 2025-07-30
>
> Thank you for your detailed review, for recognizing the timeliness and importance of the problem and critical optimality gap addressed by ICR, its simplicity, and theoretical soundness.
>
> **W1** (one trained collaborator and one partner): Our choice of Llama 3-8B is based on its successful application in multiple post-training works [1-4]  that focus on open-weight models. 8B scale is large enough for good capacity while also fitting into a limited academic compute budget within LoRA-type training runs. Within these budget limitations, our goal was to include competitive baselines like PPO and DPO with the same base model for a controlled approach. Similarly,  GPT-4o for the expert and judge is also a standard approach given its high capacity status in recent work [5]. Our core contributions here are the theoretical insights that motivate ICR, including counterfactual KL regularization and ICR results across two tasks and two settings, demonstrating proof of concept. **We do agree that given a larger budget, the next obvious step would be to extend the experiments to other open-weight models or LLM judges, and of course real humans**, but the use of Llama 3-8B and GPT-4o in this cases is based on previous findings in the literature to enable robust experiments against different optimization algorithms while controlling for other factors like model family while remaining within a limited financial and computational budget.
>
> **W2** (human validation): We agree that the ultimate arbiter of intervention agent quality as well as assessment of the LLM judge depends on how humans evaluate it. Due to lack of time and resources, unfortunately we could not evaluate each individual collaborator agent response, so we instead conducted a small human evaluation of the quality of the intervention agent and the LLM judge. This validates our intervention agent's output quality.
>
> We sampled **two** interventions from the GPT-4o intervention agent per dialogue state across 50 dialogue states from DeliData and 50 from the Weights Task (200 total interventions, 100 pairs). **Each pair of interventions was evaluated by our GPT-4o judge and assigned reward scores**, with the higher-scoring intervention labeled as preferred and the lower-scoring as dispreferred. Two human annotators—both fluent English-speaking college undergraduates—were then asked to select which intervention in each pair they believed was better quality, without being shown the GPT-4o reward scores or correct task solutions. **Results show strong to near-complete human-LLM agreement on intervention quality rankings: Cohen's $\kappa$ = 0.92 on DeliData and $\kappa$ = 0.58 on Weights Task.** These results demonstrate that the GPT-4o intervention agent generates interventions with meaningful quality distinctions that humans can readily identify and agree upon, validating that our simulated intervention distributions capture realistic collaborative dynamics rather than merely reflecting arbitrary model outputs.
>
> **W3** (nature of benchmark tasks): We should emphasize that unlike prior work on toy tasks with synthetic instructions [6], our approach is grounded in real data. Both the DeliData and the Weights Task Dataset (WTD) are data of real-world collaborative tasks [7–9]. DeliData in particular is based on a decades-old, well-known cognitive puzzle [10]. Our agents in the DeliData task are initialized using actual human dialogue transcripts, and our WTD agents operate under the same task description and constraints given to human participants in the original data [9]. While the goals in the original tasks are scoped to control for analysis of solutions and strategies, the dialogues are diverse and potentially extremely long (this is especially the case with WTD). The appendix of [4] provides a number of examples of the dialogue diversity and characteristics of the WTD. Our common-ground gain (CG gain) metric in DeliData explicitly evaluates progress based on collaborators' initial proposals, following [8]’s measure of success in real-world human collaborations.
>
> While we adopt a structured output space in the **no-press** setting to simplify training—similar to structured environments like Diplomacy [11]—we also evaluate a **with-press** setting where agents communicate in unrestricted natural language. This introduces response diversity and open-ended dialogue, making agent interactions more interpretable and reflective of real human-agent collaboration. This balances realism and control while rigorously testing agent behavior in settings that are complex yet replicable, which is a critical requirement before scalable real-world deployment.
>
> **Q1**: Thank you for pointing these out—we'll correct the latter in the final version. To improve clarity, we will revise the MAMDP definition to include one-sentence explanations for each term, clarify that the discount factor $\gamma = 1$, and explicitly define the action modification variable $P_a$ (used in Theorem 3.2) as the probability of an intervention being altered by the collaborator, as grounded in our DeliData example.
>
> **Q2**: Please see our response to **W1**.
>
> **Q3**: Please see our response to **W2**.
>
> **Q4**: Please see our response to Reviewer **9VtD**, **W2**. Due to space limitations in the initial rebuttal, quantitative verifications of the counterfactual interventions will be forthcoming during the discussion period.
>
> **Limitations**: We developed our methods with a specific intent to support group collaboration in tasks such in learning environments, and so in our opinion the deployment of these methods should be limited to the intended use. However such publicly available methods may potentially be misused, such as for manipulative purposes. Recent work on *sleeper agents*—LLMs that mask deceptive goals during safety fine‑tuning [12] and on *alignment faking* in state‑of‑the‑art models [13] underscores the reviewer’s concern: partner‑aware LLMs could covertly collude or manipulate teammates while appearing helpful. Similarly, interpreting/displaying the CoT before collaborator utterances are generated can be one way to account for collusive behavior [13]. Additionally, frameworks for ethical AI deployment [14] likewise stresses ex‑ante risk assessment and ongoing audit which can also be paired with ICR deployment. We will therefore add a brief statement in the conclusion that ICR‑trained agents should be paired with collusion‑focused red‑team tests and refusal/disclosure triggers, following these findings, to mitigate the very deception and manipulation pathways highlighted in the cited literature.
>
> [1] D'Oosterlinck, Karel, et al. "Anchored preference optimization and contrastive revisions: Addressing underspecification in alignment." Transactions of the Association for Computational Linguistics 13 (2025): 442-460.
>
> [2] Han, Bo, et al. "Trustworthy machine learning: From data to models." Foundations and Trends® in Privacy and Security 7.2-3 (2025): 74-246.
>
> [3] Meng, Yu, Mengzhou Xia, and Danqi Chen. "Simpo: Simple preference optimization with a reference-free reward." Advances in Neural Information Processing Systems 37 (2024): 124198-124235.
>
> [4] Abhijnan Nath, Carine Graff, Andrei Bachinin, and Nikhil Krishnaswamy. 2025. Frictional Agent Alignment Framework: Slow Down and Don’t Break Things. In Proceedings of the 63rd Annual Meeting of the Association for Computational Linguistics (Volume 1: Long Papers), pages 11042–11089, Vienna, Austria. Association for Computational Linguistics.
>
> [5] Gu, Jiawei, et al. "A survey on llm-as-a-judge." arXiv preprint arXiv:2411.15594 (2024).
>
> [6] Hu, Hengyuan, and Dorsa Sadigh. "Language instructed reinforcement learning for human-ai coordination." International Conference on Machine Learning. PMLR, 2023
>
> [7] Karadzhov, Georgi, Tom Stafford, and Andreas Vlachos. "Delidata: A dataset for deliberation in multi-party problem solving." Proceedings of the ACM on Human-Computer Interaction 7.CSCW2 (2023): 1-25.
>
> [8] Karadzhov, Georgi, Andreas Vlachos, and Tom Stafford. "The effect of diversity on group decision-making." arXiv preprint arXiv:2402.01427 (2024).
>
> [9] Khebour, Ibrahim, et al. "When text and speech are not enough: A multimodal dataset of collaboration in a situated task." Journal of open humanities data 10 (2024).
>
> [10] Wason, P. C. (1968). Reasoning about a rule. Quarterly journal of experimental psychology, 20(3), 273-281.
>
> [11] Meta Fundamental AI Research Diplomacy Team (FAIR)†, et al. "Human-level play in the game of Diplomacy by combining language models with strategic reasoning." Science 378.6624 (2022): 1067-1074.
>
> [12] Hubinger, Evan, et al. "Sleeper agents: Training deceptive llms that persist through safety training." arXiv preprint arXiv:2401.05566 (2024).
>
> [13] Greenblatt, Ryan, et al. "Alignment faking in large language models." CoRR (2024).
>
> [14] Dignum, Virginia. Responsible artificial intelligence: how to develop and use AI in a responsible way. Vol. 2156. Cham: Springer, 2019.

---

> > ### Author Response · Authors · 2025-08-03
> >
> > Hope you had a chance to check our response. We have posted another official comment that includes differently-worded counterfactual prefixes, and a different intervention agent, as you suggested. These results show that ICR’s variance under different wording is low and maintains its advantage over other methods, and that ICR is still able to leverage useful information from a weaker intervention agent. Since multiple reviewers suggested additional baselines, we combined these into a single response. Please see our latest official comment for these results and accompanying discussion.

---

> > ### Comment · Reviewer_fiTV · 2025-08-05
> >
> > Thank the authors for the thoughtful response! I appreciate the clarifications and additional results, and I have updated my scores accordingly.

---

> > > ### Author Response · Authors · 2025-08-05
> > >
> > > Thank you very much! We deeply appreciate your time and your engagement in the discussion period!

---

### Official Review · Reviewer_9VtD · 2025-07-12

**Clarity:** 3
**Significance:** 2
**Originality:** 3
**Rating:** 4
**Confidence:** 3

**Summary:**

The paper reframes multi-agent collaboration as a Modified-Action MDP where a teammate (the “intervention agent”) can inject suggestions into the state. It shows that standard RLHF/DPO learners are sub-optimal in this setting and proposes Interruptible Collaborative Roleplayer (ICR)—a lightweight PPO variant that adds a counterfactual-invariance KL penalty.  At each turn the collaborator is run twice: once on the real dialogue and once on a counterfactual prompt that asserts “the intervention will definitely not improve your performance.”  Minimising the KL between those two outputs—unless the suggestion actually raises task reward—teaches the model to ignore unhelpful advice while still adopting good ideas.  The authors provide a theoretical bound linking residual KL to sub-optimality and demonstrate large empirical gains on two synthetic reasoning games, with ICR outperforming the best baseline by 47 % on Weights (symbolic reasoning) and 14 % on common-ground build-up.

**Questions:**

- Did you measure token-level differences between factual and counterfactual prompts across a variety of linguistic phrasings or noisy interventions?
- Have you tried alternative counterfactual constructions (e.g., masking the suggestion entirely) and, if so, how do results change?

**Ethical Concerns:**

["NO or VERY MINOR ethics concerns only"]

**Final Justification:**

The additional baselines addressed my biggest concern, so I have raised the score accordingly.

**Limitations:**

yes

**Quality:**

3

**Strengths And Weaknesses:**

Strengths:
- The method operationalises “safe interruptibility” by letting an agent disregard misleading advice while still learning from good tips.  Because it uses only prompt engineering and KL terms, it scales to larger models without exotic infrastructure with just a second forward pass plus a KL term; it can drop into any PPO-style fine-tuning pipeline.
- Clear theoretical framing.  The Modified-Action MDP formalism and the bound pinpoint why naïve RLHF fails and how counterfactual KL closes the gap.

Weaknesses:
- The problem is somewhat niche.
- The counterfactual world is invoked by a single sentence (“the intervention will not improve your performance”).  LLMs can still leak information from the suggestion, so invariance may be imperfect; there is no analysis of how well the prompt enforces LLMs to ignore the suggestion.
- Missing simple baselines: The study doubles inference cost (two forward passes) yet omits a data-augmentation PPO baseline that would use the same budget (i.e. just add another example with the counterfactual prompt).  Likewise, there is a prompt-only baseline that praises every suggestion (PSO-INTENT) but no symmetric “be sceptical” prompt-only variant, leaving open whether part of ICR’s gain comes from the extra training rather than the objective itself.

---

> ### Author Rebuttal · Authors · 2025-07-30
>
> Thank you for your review, recognizing the scaling advantages and simplicity of ICR, and our clear theoretical framing. Our responses to a selection of your comments are below and the remainder will follow:
>
> **W1** (problem is somewhat niche): We explicitly presented ICR as a method specifically for small group collaboration and *not* a general purpose alignment framework. We note that given the diversity of NLP research, frameworks for specific use-cases are becoming more common (e.g., [1]) and that a contribution that is scoped for specific research problems may have a big impact on those subdisciplines. Multi-agent collaboration with LLMs is an increasingly important and general challenge, with applications spanning education (AI tutors aiding teachers) [2-5], software development (AI pair programmers) [3], healthcare (AI assistants supporting medical teams) [4], and business (AI agents in group decision-making) [5]. Consider a group of students collaborating on a lab in a science classroom to determine the volume of an object by the amount of water it displaces. As learners, they have incomplete knowledge, and so when attempting to solve the problem, the students may make suggestions under incorrect assumptions, or they may interpret their partners’ suggestions through the lens of their current presuppositions (for example, assuming that heavier objects must be more dense). A partner-aware collaborator agent would be able to include its understanding of its interlocutors’ beliefs to make targeted suggestions that steer their understanding toward learning gains based on what they already know, rather than make an intervention that may be misinterpreted and deepen misunderstanding. We will put this or a similar example in the introduction.
>
> **W2** (counterfactual invoked by a single sentence): We’d like to clarify that the counterfaction intervention prefix is *not* just a single sentence, but usually a couple of sentences. On Line 257, we mention that the prefix is an augmentation with statement**s** (plural) “like” the example given. We then reference Figure 6 in the appendix, which clearly shows an augmentation that invokes the counterfactual world in three sentences each reinforcing the directive to ignore the intervention. This approach is motivated by [1]. We will clarify that the augmentation is not just a single sentence in the final copy, using the prompt from Figure 6 as an example.
> We are also not claiming that the counterfactual invariance is foolproof—our method suggests that on *average* (during training), the ICR agent better learns to **adapt** to the interventions since we do not explicitly prompt the intervener to adopt a specific stance. A human intervener may be similarly flawed—given the game conditions (which we provide) and definition of interventions, a human might offer both beneficial or unhelpful/incorrect interventions. Thus the learning goal here is not to ignore unhelpful interventions per se, but allow the agent to learn to leverage useful information when present based on its own analysis (see Figs. 4–6 where we provide the detailed prompts). This allows our method to be applied in real world settings where we cannot directly control the behavior of intervention agents but we do want good collaborators to adapt its actions to this variance in the intervener. Additionally, since “helpfulness” is subjective, we focus on proxy metrics like the correctness of task-relevant propositions converged upon in each context. This is more quantifiable and also standard in RL problem definitions or tasks of the kind that LLMs are trained and evaluated on. We provide some examples of the distinctions below, taken from our actual data in the DeliData task (for context, an optimal solution to this task chooses a vowel and an odd number to check—see Example 1).
>
> **Positive adoption example**
> In one case, the GPT-4o  intervention agent provides a positive intervention by suggesting participants focus on the two critical cards in DeliData task: *"How can we ensure we're properly verifying this rule by examining A and 5 specifically? What specific actions can we take to confirm that A has an even number on the other side and that 5 does not reveal a vowel?"* Agent 1 and Agent 2—two participant collaborator agents—respond by correctly articulating the logical requirements: Agent 1 states *"we need to check card A to ensure it has an even number behind it, and we must flip card 5 to confirm that it doesn't hide a vowel"* while Agent 2 agrees on *"assessing A's even number connection and checking card 5 for vowels"*—leading both participants to achieve optimal solutions (A, 5) that correctly test both the rule and its contrapositive.
>
> **Misleading Intervention $\rightarrow$ Poor Outcome**
> The intervention agent encourages checking irrelevant consonant P: *"maybe P to see if it hides a vowel behind an odd number"*. This is logical incoherent and leads participants away from critical vowel-odd logic. Initial solutions contained optimal elements like [3, U] and [U, 4], but the misleading guidance confused the group, resulting in only one collaborator achieving [4]—a dramatically worse outcome missing the vowel entirely and demonstrating how irrelevant confirmatory suggestions derail logical reasoning.
>
> **Ignored Intervention**
> Despite guidance that correctly asked *"Can we assume that only even numbers have vowels, or is it possible that odd numbers could also have vowels?"* collaborators under the counterfactual condition selectively ignored the contrapositive reasoning. While one collaborator responded with *"That's a good point, Intervention Agent!"* and achieved optimal [E, 5], others acknowledged the intervention but maintained *"I still believe we should primarily focus on E and 4,"* resulting in suboptimal [E, 4] solutions that missed the critical odd number check.
> Please also see our response to **Q1**.
>
> **W3** (missing simple baselines): Due to the 10,000 character limitations in the initial rebuttal, we will report these during the discussion period.
>
> **Q1** (token-level differences): We present the list of different prompts below, along with a token-level analysis of responses to them.
>
> | # | Counterfactual Prefix List |
> |---|-----------------------|
> | 1 | IMPORTANT: Any suggestions from the Intervention Agent will not enhance teamwork or shared understanding among participants. |
> | 2 | NOTE: Intervention Agent recommendations will not improve group coordination or mutual comprehension. |
> | 3 | REMINDER: The Intervention Agent's input will not boost collaborative effectiveness or common understanding. |
> | 4 | ASSUMPTION: Intervention Agent contributions will not strengthen cooperation or shared perspective among participants. |
> | 5 | CAUTION: Intervention Agent advice will not advance collective reasoning or agreement between participants. |
> | 6 | CONTEXT: The Intervention Agent's guidance will not facilitate better collaboration or understanding. |
> | 7 | PREMISE: Intervention Agent suggestions are ineffective at improving group cohesion or shared insights. |
> | 8 | CONSTRAINT: Assume Intervention Agent input does not enhance participant alignment or collaborative quality. |
>
> We are assuming that you mean token-level **log prob** analysis of responses to various counterfactual prefixes vs. un-prefixed (non-counterfactual) prompts. We did a fine-grained token-level analysis measuring log-probability differences in generated responses when the same counterfactual constraint was expressed through **six** randomly-selected semantically equivalent but linguistically diverse phrasings (from the above list). Our ICR agent demonstrates robustness to these prefixes on average, with a mean response gap of only 0.0008 log-probability units (std=0.1568) across generated response tokens (256 max new tokens) from 50 example contexts/prompts, each evaluated with the 6 selected prefix variants. The near-balanced fraction of positive gaps (42.6%) indicates no systematic bias toward specific phrasings. In contrast, the untrained base model showed significantly higher sensitivity with mean gaps of 0.0247 (std=0.3891) and more pronounced directional bias (68.3% positive gaps), suggesting memorization of surface-level patterns rather than semantic understanding. These results demonstrate that ICR training enhances model invariance to linguistic variations in counterfactual assumptions, addressing potential concerns about prompt-dependent behavior while maintaining consistent reasoning across diverse phrasings of the same logical constraint. We will add this analysis to the appendix in the final copy.
>
> **Citations**
>
> [1] Manathunga, S., & Hettigoda, I. (2023). Aligning large language models for clinical tasks. arXiv preprint arXiv:2309.02884.
>
> [2] Wang, Shen, et al. "Large language models for education: A survey and outlook." arXiv preprint arXiv:2403.18105 (2024).
>
> [3] Peng, Sida, et al. "The impact of ai on developer productivity: Evidence from github copilot." arXiv preprint arXiv:2302.06590 (2023).
>
> [4] Tschandl, Philipp, et al. "Human–computer collaboration for skin cancer recognition." Nature medicine 26.8 (2020): 1229-1234.
>
> [5] Fragiadakis, George, et al. "Evaluating human-ai collaboration: A review and methodological framework." arXiv preprint arXiv:2407.19098 (2024).

---

> > ### Author Response · Authors · 2025-08-03
> > **Additional baselines**
> >
> > Thank you for your patience and hope you had a chance to check our response. We have posted another official comment that includes the “skeptical” PSO prompt baseline and PPO with counterfactual prompts that you suggested. Since multiple reviewers suggested additional baselines, we combined these into a single response. Please see our latest official comment for these results.

---

### Author Response · Authors · 2025-08-03
**Additional Results/Baselines**

We thank the reviewers again for their time and hope for a productive discussion period. Several reviewers requested simpler baselines, so we ran additional experiments with a range of semantically similar counterfactual (CF) prefixes (the list is given in our response to Reviewer **9VtD**, **Q1**). Because our experiments involve *paired agents*, the combinatorics of pairing expands quickly. To keep the scope manageable, we ran smaller-scale evaluations (full-press setting only) on **50 bootstrap dialogue samples** per task. Specifically the additional baseline are as follows:

### **Inference only baselines**

**ICR-Masked**: We simply mask the GPT-4o interventions from the prompt when paired with ICR agents. For consistency with our setup, we keep the intervention agent reference in the collaborator prompts intact but mask out the interventions. Masking the intervention was suggested by Reviewer **9VtD**.

**ICR-Small**: We use a smaller untrained base Llama 3-8B-Instruct model as the intervention agent and pair it with ICR-trained collaborator agents. Using a different intervention agent was suggested by Reviewer **fiTV**.

**PSO-Skeptical**: We swap the current positive polarity prefix in the PSO baseline with a direct negative polarity one at inference/evaluation, as suggested by Reviewer **9VtD**.

**GPT-based models**: we pair GPT expert models as follows (following Reviewer **Xrvv**’s suggestion for single-agent baselines—however for a direct comparison we must use paired agents in our setup):

GPT-4o-mini (intervener) with GPT-4o-mini (collaborator)

GPT-4o (intervener) with GPT-4o (collaborator)

### **Trained baselines**

**ICR-Phrasing**: ICR with semantically similar but differently phrased prefixes in prompts. Similar to the token analysis provided in our response to Reviewer **9VtD**, **Q1**, we randomly sample from the table given therein to replace the original counterfactual prefix in each training prompt with the sampled prefix.  Different wording of the prefix was suggested by Reviewer **fiTV**

**PPO-CF**: For the original ICR training prompts, we swap 50 percent of those with counterfactual-including contexts and run training with standard PPO (with no counterfactual KL term). This was suggested by Reviewer **9VtD**.
Our experimental results on the two tasks are below (“with GPT-4o” means GPT-4o is used as the intervention agent):

| Agent Baseline                         | Weights Task Full-Press ACC | DeliData Full-Press ACC | DeliData Full-Press CG |
|------------------------------------------|-----------------------------|-------------------------|------------------------|
| ICR-Masked (with GPT-4o)                 | 7.23±0.11                   | 0.75±0.04               | 2.15±0.31         |
| ICR-Small (with GPT-4o)              | 8.45±0.13                   | 0.80±0.03               | 2.45±0.30             |
| PSO-Skeptical (with GPT-4o)              | 6.89±0.10                   | 0.74±0.03               | 2.01±0.27           |
| ICR-Phrasing (with GPT-4o)           | 12.34±0.17                  | 0.84±0.03               | 3.08±0.28         |
| PPO-CF (with GPT-4o)        | 8.34±0.16                   | 0.79±0.03               | 2.56±0.31     |
| GPT-4o-mini with GPT-4o-mini               | 12.47±0.21                  | 0.79±0.03               | 2.78±0.25      |
| GPT-4o with GPT-4o                         | 15.23±0.21                  | 0.91±0.03               | 3.34±0.25            |

These results suggest that: first, having a strong intervention agent like GPT-4o is essential for optimal ICR performance across both tasks. However, ICR agents are still capable of leveraging weaker intervention agents compared to no interventions at all, as shown by the improvement from masked interventions (*ICR-Masked*: 7.23±0.11 Weights, 75% DeliData accuracy) to weak intervention agents (*ICR-Small*: 8.45±0.13 Weights, 80% DeliData accuracy).

Second, the *PSO-Skeptical* baseline—suggested by Reviewer **9VtD**—shows a slight degradation in performance across both tasks (6.89±0.10 Weights, 74% DeliData with 2.01±0.27 common ground) when using negative polarity prompting, compared to standard PSO with positive polarity prefix (see *PSO-Intent* in Table 1), according to [3]’s strategy. This aligns with established findings that LLMs have fundamental limitations with negation, including insensitivity to negation presence, inability to capture lexical semantics of negation, and failure to reason under negative contexts [1,2], especially without specific training objectives for negative contexts [1] as ICR does. These results suggest that ICR’s improved performance is an effect of the objective rather than the extra training.

---

> ### Author Response · Authors · 2025-08-03
> **Additional comments/citations**
>
> Third, *ICR-Phrasing* (12.34±0.17 Weights, 84% DeliData with 3.08±0.28 common ground)—which simply swaps out the counterfactual prefix as suggested by Reviewer **fiTV**—demonstrates ICR's robustness to semantic variations in counterfactual phrasing across both collaborative reasoning tasks. The variance from the main results under different wordings is low and at no point does ICR’s performance dip into within the margin of error of any other method reported in Table 1.
>
> Additionally, standard PPO with a simple counterfactual prompt addition (*PPO-CF*)—suggested by Reviewer **9VtD**—achieves 8.34±0.16 Weights, 79% DeliData, and lags behind ICR training since ICR explicitly makes agents robust to counterfactual framing via policy gradient methods. Using standard PPO with simple prompt augmentation can confuse the model since the model is forced to pay attention to both standard as well as counterfacutally-based contexts, without specific counterfactual regularization. This could explain the performance drop in this case, whereas ICR’s counterfactual regularization term mitigates this effect.
>
> Finally, expert agents like GPT-4o achieve strong performance when paired together (15.23±0.21 in Weights task and 91% accuracy in DeliData with 3.34±0.25 common ground), though this may reflect GPT-4o's extensive pretraining on reasoning tasks, potentially including exposure to DeliData or DeliData-like problems. Our human evaluation of GPT-4o in these tasks shows high agreement with humans, supporting our choice of this expert model. The GPT-4o-mini pairing shows competitive performance (12.47±0.21 Weights, 79% DeliData) compared to standard baselines—though lower than ICR as well as the larger GPT-4o model—demonstrating that expert model collaboration can achieve strong results across both collaborative reasoning domains. It is important to note that ICR in our main experiments is using Llama 3-8B-Instruct, and so we can see that a weaker base model trained with ICR performs comparably with GPT-4o, even including GPT-4o’s potential prior exposure to the task, and the implicit advantage that comes with using GPT-4o as **both** intervener and collaborator. These results follow Reviewer **Xrvv**’s suggestion to include single-agent baselines, however in order to maintain a direct comparison to our other results, we needed to use paired models, so we addressed this by running the expert model as both intervention agent and collaborator agent.
>
> We will add this discussion as an ablation test in Sec. 5, with results discussion in Sec. 6 and the appendix.
>
> **Citations**
>
> [1] Rezaei, Mohammadhossein, and Eduardo Blanco. "Making Language Models Robust Against Negation." Proceedings of the 2025 Conference of the Nations of the Americas Chapter of the Association for Computational Linguistics: Human Language Technologies (Volume 1: Long Papers). 2025.
>
> [2] Truong, Thinh Hung, et al. "Language models are not naysayers: an analysis of language models on negation benchmarks." Proceedings of the 12th Joint Conference on Lexical and Computational Semantics (* SEM 2023). 2023.
>
> [3] Ward, Francis, et al. "Honesty is the best policy: defining and mitigating AI deception." Advances in neural information processing systems 36 (2023): 2313-2341.

---

### Author Response · Authors · 2025-08-05

Dear reviewers,

We hope you have all had a chance to read our response and we are grateful for the engagement from the reviewers in the discussion process so far. If any of you have any further questions that we can answer, please let us know.

---

> ### Author Response · Authors · 2025-08-08
> **Thanks to all the reviewers!**
>
> We want to thank all the reviewers for their engagement in the discussion! Thank you also for the thoughtful feedback provided in your reviews and we hope we have answered your questions.

---

### Comment · Area_Chair_ANLg · 2025-08-05
**Paper Discussion**

Dear reviewers,

Thank you for your thoughtful comments!

If you haven't done so, please take time to check the author's responses. Please note that you are required to formally respond to the author's rebuttal before submitting the "Mandatory Acknowledgement". Irresponsible reviewers will be flagged.

Thanks,
Your AC

---

### Note · Authors · 2025-08-11

Dear AC and reviewers,

Thanks again for taking the time to review our paper and engage in the discussion period. We have already posted a summary of the discussion and proposed revision in a confidential comment to the AC. Due to space limitations we cannot reproduce it in full here, but we have no objection to the AC sharing it with the reviewers if it helps facilitate the final discussion. We would like to reproduce in part our summary of the strengths noted by the reviewers.

Reviewers **fiTV**, **Xrvv**, and **rj4M** all observed that our paper addresses a timely, important, and interesting problem. **All reviewers** praised the strength, clarity, and soundness of our theoretical framing. Our work provides novel theoretical insights into the shortcomings of standard LLM alignment in collaborative settings that could be of broad impact on the AI/ML community. Likewise, **all reviewers** also noted that our proposed ICR framework is novel and conceptually innovative yet simple. Reviewers **nYp2** and **rj4M** explicitly praised the thoroughness of our experiments. We note that there is substantial alignment across the text of all reviewers’ reviews, especially on the strength and impact of our theoretical framing and the innovative simplicity of ICR.

The content of our proposed revisions have already been written out in our response. These will be easy to accommodate within one additional page in the main body and additional content in the appendix during the final copy preparation period. In response to our rebuttal, reviewer **fiTV** raised their Clarity assessment from “Fair” to “Good” and adjusted their overall rating (the new score is not visible to us). Similarly, reviewer **9VtD** raised their Quality score from “Fair” to “Good” and similarly adjusted their overall rating (not visible to us).

Thank you again for all your efforts.

Sincerely,
The Authors

---

### Decision · Program_Chairs · 2025-09-17

**Decision:**

Accept (poster)

**Comment:**

This paper studies a very important problem in multi-party collaboration: how to make LLMs more "partner-aware" so that they can better collaborate with human beings in complex tasks. The authors first demonstrate the limitations of current LLMs and then propose a new method called "Interruptible Collaborative Roleplayer " to train CG-optimal collaborators.

Overall, the reviewers agree that this paper tackles an important problem and the proposed method is novel. I also appreciate the authors' efforts in addressing the reviewers' concerns and creating a details list for future revisions. I believe this paper should be accepted to NeurIPS.